# Recent Advances in Optical Hydrogen Sensor including Use of Metal and Metal Alloys: A Review

**Akhilesh Kumar Pathak** [1][iD], **Sneha Verma** [2][iD], **Natsima Sakda** [3], **Charusluk Viphavakit** [1][iD], **Ratchapak Chitaree** [3][iD] **and B. M. Azizur Rahman** [2,*][iD]

1   International School of Engineering and Intelligent Control Automation of Process Systems Research Unit, Faculty of Engineering, Chulalongkorn University, Bangkok 10330, Thailand
2   School of Science and Technology, City University of London, London EC1V 0HB, UK
3   Department of Physics, Faculty of Science, Mahidol University, Bangkok 10400, Thailand
*   Correspondence: b.m.a.rahman@city.ac.uk

**Abstract:** Optical sensing technologies for hydrogen monitoring are of increasing importance in connection with the development and expanded use of hydrogen and for transition to the hydrogen economy. The past decades have witnessed a rapid development of optical sensors for hydrogen monitoring due to their excellent features of being immune to electromagnetic interference, highly sensitive, and widely applicable to a broad range of applications including gas sensing at the sub-ppm range. However, the selection of hydrogen selective metal and metal alloy plays an important role. Considering the major advancements in the field of optical sensing technologies, this review aims to provide an overview of the recent progress in hydrogen monitoring. Additionally, this review highlights the sensing principles, advantages, limitations, and future development.

**Keywords:** dielectric; waveguide; sensors; hydrogen; detection; metal alloy; hydrogen economy; gas sensor

## 1. Introduction

A demand for economic growth also leads to an even larger energy consumption demand. At present, the fossil-dominated world energy structure, however, has brought about increasingly serious environmental problems, by releasing greenhouse gases causing global warming, acid rain, and ozone layer destruction. This leads to an urgent need to find new abundant clean energy resources. Hydrogen gas is considered as an alternative renewable energy source to replace the traditional fossil fuel-based energy sources to avoid air pollution, carbon emissions, and climate change [1,2]. Hydrogen is now being utilized in several industrial sectors in order to produce green energy and power for several applications [3,4]. Excess solar energy available in the day can be used to produce green hydrogen which can be used to produce electricity in night. Considering the wide application of hydrogen, it is important to make the technology safer for the public. More than 156 countries unveiled their new 'Hydrogen Strategies' in Glasgow in 2021, and prepared plans to transform their energy usage and infrastructure to become 'Carbon-Zero' by 2030 [5]. The success of hydrogen strategies depends on the generation, storage, transportation, and wider uses of hydrogen.

Hydrogen has a high heat of combustion (142 kJ g$^{-1}$), a wide range of explosive concentration (4–75 vol%), low ignition energy (0.02 mJ), and high flame propagation velocity [6]. Therefore, regular monitoring of its concentration and detection of leakage is highly desirable to give the public the confidence in using this technology routinely. The concentration monitoring of hydrogen has a long history of over 100 years since the introduction of hydrogen measurements for airships at filling stations. The major use of hydrogen detection is important in the hydration of hydrocarbons, synthesis of methanol and ammonia, the desulphurization of petroleum products, and the production of fuels for the rocket. The real-time monitoring of hydrogen leakage is also highly required for

nuclear reactor safety. In the nuclear power stations, the hydrogen gases are produced in the radioactive waste tanks during plutonium reprocessing via the unwanted reaction of water with high-temperature reactor cladding and core materials (zirconium, uranium oxide). In coal mines, hydrogen can be produced in the very low concentration of ppm range by coal-dust explosions, methane, or the low-temperature oxidation of coal [7]. In addition to the industrial sector, the hydrogen sensor also plays a vital role in medical applications as a biomarker and for the monitoring of environmental pollution [7]. The recent growth in hydrogen technology evidently shows its future in household appliances and car filling stations. In future we can expect to have $H_2$ alone or mixed with natural gases for cooking or central heating; therefore, hydrogen safety at home and at car filling station is required to avoid any incidents. Hydrogen is also known to be released from the ocean by submarine hydrothermal activity. The dissolved hydrogen has been shown to play an important role in a variety of biological process in the marine environment. For example, hydrogen can be produced and consumed by several aerobic aquatic organisms, and it is an important intermediate in the anaerobic oxidation of organic matter [8–10]. In addition to the marine environment, hydrogen is also necessary in aquaculture. When several fish farms do not have access to shore power, 4 out of 10 farms can run on diesel generators which produce $CO_2$ [11]. To fulfill their requirements approximately 10 tons of green hydrogen per plant need to be produced. The coastal aquaculture industry is considered as the most predominant form of aquaculture but has caused numerous conflicts among stakeholders because of the limited availability of power in near-coast areas combined with relevant environmental issues [12]. The latter combined with the additional need of power for seafood production has led to a growing trend for the industry to move further away from coast. Sustainable and competitive offshore economic activity though faces many challenges. A critical one is related to energy requirements and how to provide this vital element uninterruptedly far from the coast. The hydrogen energy can play a crucial role for supplying energy offshore. The requirement of hydrogen on a large scale required a strategic need for better means of in situ hydrogen monitoring. This drives the need for real time, low-cost, sensitive, and reliable hydrogen leakage detection to give the public the confidence in using this technology routinely and more widely. Ongoing research, development, and as yet small-scale deployment of hydrogen technologies seek to realize these potentials. Therefore, in this emerging hydrogen economy, hydrogen sensing devices can be used in wide applications of safety monitoring such as at hydrogen production plants, storage tanks, pipelines, homes, refueling stations, and automotive vehicles.

Conventionally hydrogen sensing is performed by a gas chromatography mass spectrometer (CGMS) or specific ionization gas pressure sensors. These techniques utilize the similar method of ion-pair extraction of the gas and quantification by MS and may detect the concentrations as low as ppm [13]. However, the most significant limitations of these techniques are typically very time consuming, expensive, require skilled technicians, off-site analyses, and matrix-matched calibration standards that are not routinely determined [14]. To overcome these issues, various other techniques have been considered to detect hydrogen, e.g., electrochemical [15], catalyst [16], resistance based [17], and optical methods [18–20]. However, electrochemical and optical based sensors are the most preferred because they are capable of detecting low concentrations of hydrogen with an acceptable selectivity [3,21]. Among these technologies, the metal oxide such as tungsten ($WO_3$) gas sensors is considered one of the popular materials for device fabrication due to their wide energy band gap. However, the chemical reaction between hydrogen and $WO_3$ is not strong enough [22]. Therefore, most of the hydrogen sensor utilizes palladium (Pd) in hydrogen sensing due to its selective absorbance of hydrogen and its ability to work at room temperature [23]. The expansion of its lattice constant changes its refractive index, density, and introduces stress. It also changes acoustic wave propagation through a Pd surface layer [24]. Hydrogen sensing, either by optical method or surface acoustic wave (SAW) based approaches, selection of Pd, and Pd-alloy plays a key role. Sensing parameters including selectivity, sensitivity, and operating temperature can be improved by alloying

Pd with noble metals such as gold (Au), copper (Cu), silver (Ag), platinum (Pt) [25], etc. An earlier comprehensive review of these sensing technologies can be found in a paper by Hübert et al. [7]. Additionally, a report of gas sensing technology with optochemical techniques was also summarized by Ando in 2006 [26]. A summary on optical fiber based hydrogen sensors is shown in Figure 1.

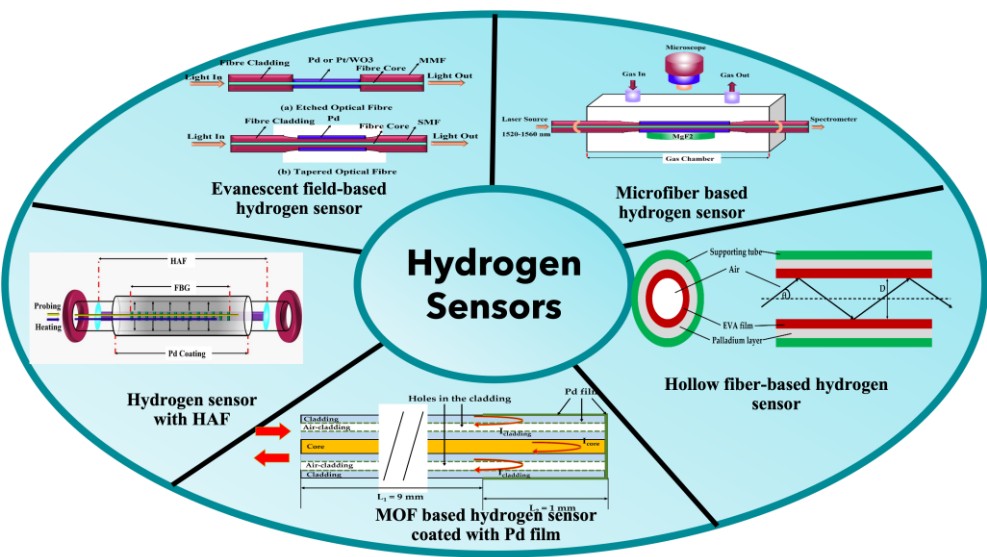

**Figure 1.** Summary of all optical fiber based hydrogen sensors.

Several types of hydrogen sensors are commercially available on the market. Most of the hydrogen sensing principles have been known for decades. Although to meet the requirements of a future hydrogen economy, a lot of research is ongoing in order to continuously improve the sensing response including its selectivity, sensitivity, and response time in addition to reducing the sensor size, cost, and power consumption. The requirement of hydrogen sensing technology can be found in previously published articles [7,27,28]. The increase in commercial interest and R&D due to the emergence of new and widespread expected applications for these sensors is reflected in the growing number of relevant publications since the year 1970 as shown in Figure 2.

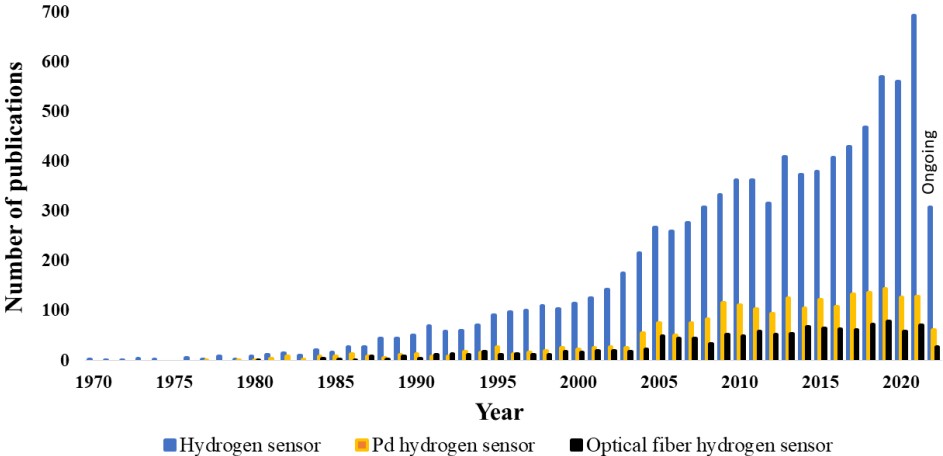

**Figure 2.** Growth in the publication of hydrogen sensors, Pd hydrogen sensors, and optical fiber hydrogen sensors since 1970 according to SCOPUS (June 2022).

This review provides a detailed overview on the development of optical sensing technology using metal hydride as follows. In Section 2, a brief discussion on sensing materials and sensing principles are discussed. Section 3 comprises the discussion on various

strategies and measurement methods used to monitor hydrogen concentrations. The sub-sections contain a detailed discussion on various optical sensing technology reported so far. Section 4 contains the information on other novel and promising hydrogen sensors reported in the past few years including SAW, stress, micromirrors, metamaterials, nanoantenna, etc. Section 5 discusses the future prospects and challenges for the development of optical hydrogen sensors. Finally, an overall conclusion is drawn in Section 6.

## 2. Sensing Materials

Some materials such as palladium (Pd), tungsten oxide ($WO_3$), magnesium (Mg), and yttrium (Y) demonstrate the excellent capability to absorb hydrogen gases. The permittivity of the metal is influenced by the insertion of hydrogen atoms into the metal lattice and its volume expansion, which is monitored to determine the concentration of hydrogen in ambient environment [29]. Generally, the main developing trends of hydrogen sensors are focused on two sensitive materials which are (a) tungsten oxide ($WO_3$), and (b) palladium (Pd), which are highly selective and sensitive for hydrogen compared to any other materials [15].

### 2.1. Tungsten Oxide (WO₃)

$WO_3$ is popularly known as a semiconductor metal oxide with a perovskite-like atomic architecture. It comprises a wide bandgap varying from 2.60 eV to 3.25 eV [30,31]. The change in chemical properties of $WO_3$ in presence of hydrogen has already been reported in various literatures. The interaction between $WO_3$ hydrogen causes a change in color from greenish yellow to blue due to an exothermic effect. The chemical reaction can be defined as follows [32]:

$$xH_2 + WO_3 \leftrightarrow WO_{3-x}.xH_2O \tag{1}$$

The reaction is completely reversible leading to the increase of absorption of hydrogen in the visible range. Additionally, the interaction between $WO_3$ film and hydrogen also modulates its optical properties including its absorption, reflectance, transmittance, and refractive index, which makes it an excellent sensing material for hydrogen monitoring. However, it has also been observed and reported by several researchers that the chemical interaction between $WO_3$ and the hydrogen is not strong enough. The $WO_3$ film also reacts chemically with other gases available in mix gaseous environment e.g., acetylene and hydrogen sulfide, etc. Therefore, the cross sensitivity and inaccurate selectivity for hydrogen can occur. The optical fiber hydrogen sensor based on $WO_3$ film owns excellent absorbance and rapid response, however its performances degraded due to the presence of oxygen in the ambient atmosphere. Additionally, the sensors based on $WO_3$ are not considered to be suitable for some special requirements, such as a nuclear waste tank. Besides, in the process of exothermic reaction, the ambient temperature can be increased to hundreds of Celsius, and under 4% hydrogen concentration will have increased risk of hydrogen explosion. Therefore, the sensors based on $WO_3$ are limited for variety of applications.

### 2.2. Palladium (Pd)

Besides $WO_3$, Pd has attracted huge attention in hydrogen sensing since its potential was first demonstrated in 1961 [33]. When hydrogen gas interacts with Pd film, the hydrogen molecule gets dissociated into their atomic hydrogen, leading to its lattice expansion, which is later utilized to determine the hydrogen concentration. Additionally, the interaction of Pd with hydrogen also leads to the structural change of Pd that induces a change in its electrical, mechanical, as well as its optical properties. In 2004, Fedtke et al. reported a detailed study on the hydrogen sensor that performs as the optical switching between the optical and electrical properties of Pd thin film [34]. The basic principle of hydrogen gas sensing with the Pd layer is the phase transition between alpha ($\alpha$) and beta ($\beta$) from the

absorption of hydrogen by Pd. The absorption of hydrogen by the palladium layer and the resulting formation of the compound can be described by the following reaction [35].

$$(\alpha)Pd = \frac{k}{2} H_2 \leftrightarrow (\beta)PdH_k \tag{2}$$

the effect of the absorption of hydrogen gas on the complex dielectric function of palladium can be represented by the following equation,

$$\varepsilon_{Pd} = h(c).\varepsilon_{Pd}(0) \tag{3}$$

where, $\varepsilon_{Pd}$ is the dielectric function of the palladium layer in the presence of concentration $c$ of hydrogen gas, whereas $\varepsilon_{Pd}(0)$ is its dielectric function in the absence of hydrogen gas. The variable $h(c)$ represents a nonlinear function that decreases with the increase in the concentration of hydrogen gas. Its values for 0% and 4% hydrogen are $h(0) = 1.0$ and $h(4) = 0.8$, respectively. The Equation (2) can be used to determine the permittivity of Pd exposed to a 4% hydrogen concentration. However, there are some specific limitations of the equation which fail to address (i) the wavelength dependent nature of the hydrogenation effect on permittivity; and (ii) the different effect of hydrogenation on the real and imaginary parts of the permittivity. These limitations were resolved by Perrotton et al. in 2011 [36] and later modified by Downes et al. in 2017 [37]. Downes et al. determined the dielectric permittivity of PdH by fitting a 6th order polynomial to experimental data generated by Rottkay and Mubin [38]. The polynomial fit covered a broad wavelength range varying from 300 nm to 2000 nm. The dielectric permittivity of pure palladium $\varepsilon_{Pd}$ to that of palladium in the presence of 4% hydrogen $\varepsilon_{PdH}$ can be represented by following relation,

$$\varepsilon_{PdH} = h_1.\varepsilon'_{Pd} + ih_2.\varepsilon''_{Pd} \tag{4}$$

$$h_1 = \sum_{j=0}^{6}\lambda^j a_j \tag{5}$$

$$h_2 = \sum_{j=0}^{6}\lambda^j b_j \tag{6}$$

where a and b are the Bessel's functions.

When the Pd layer undergoes a volumetric expansion during hydrogen detection, due to the formation of palladium hydride, and changes the output wavelength with the concentration of hydrogen. Among several sensing materials, Pd has been exploited widely to develop highly selective and fast responding hydrogen sensors. In the presence of hydrogen, the Pd absorbs hydrogen molecules and breaks them into atoms, which subsequently modulate its physical and mechanical properties due to the lattice expansion of Pd as shown in Figure 3 [23].

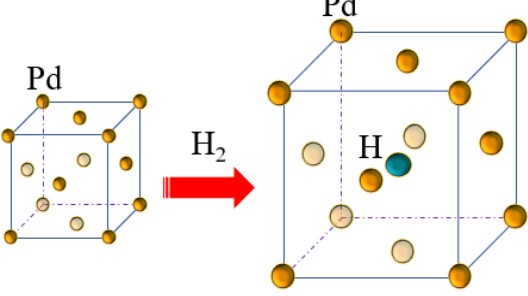

**Figure 3.** Expansion of the Pd lattice after hydrogen absorption.

## 2.3. Advantage and Limitations

Although both materials show excellent sensing performance towards hydrogen concentration, each material comprises advantages and limitations. A detailed comparative

discussion of the advantages and limitations of these hydrogen sensing materials is provided in Table 1.

**Table 1.** Advantage and limitations of $WO_3$ and Pd based hydrogen sensor.

| Sensing Material | Advantage | Limitations | Commercialization |
|---|---|---|---|
| $WO_3$ | • The $WO_3$ based hydrogen sensors work on the changes in their optical properties and color variation, visible to eyes.<br>• Can work at low or room temperatures.<br>• Excellent absorbance and rapid response time. | • $WO_3$ sensors show slow coloring or bleaching times<br>• The chemical interaction between $WO_3$ and hydrogen is not strong enough.<br>• Performance degraded in presence of oxygen.<br>• Risk of explosion due to exothermic reaction. | • The devices based on $WO_3$ show poor performance for continuous monitoring.<br>• Poor selectivity as it chemically interacts with surrounding gases.<br>• Due to exothermic reaction, the material is limited to few applications only such as nuclear waste tank. |
|  | • Pd based hydrogen sensor undergoes lattice expansion and changes its refractive index, density, stress, and acoustic wave propagation.<br>• Can operate in high and low temperature.<br>• Excellent absorbance and rapid response time. | • Pure Pd based hydrogen sensor suffers embrittlement effect arises due embrittlement effect to repeated desorption of hydrogen.<br>• Pure Pd based sensor shows hysteresis.<br>• Cross-sensitivity and deactivation by poisoning gases such as CO, NO, etc. | • High selectivity for hydrogen.<br>• No exothermic reaction takes place during hydrogen sensing hence can be applied for various applications.<br>• Alloying of Pd with noble material improved the performance in terms of embrittlement and hysteresis. |

## 3. Measurement Method

Considering the property of materials, it can be observed that hydrogen sensing incorporating Pd-alloys is more selective and promising compared to $WO_3$ [39]. The absorption of hydrogen leads to changes in various physical and chemical properties of Pd including its refractive index, stress, density, and changes in acoustic wave propagation through the Pd sensing layer [24]. The deposition of Pd on the surface of fiber leads to the change in the optical signals corresponding to hydrogen concentrations [40]. Hence, the hydrogen concentration can be monitored in real-time at a long distance by measuring the output signals. At the beginning of technology, the Pd thin film was demonstrated as a potential sensing material for hydrogen leakage detection. However, the material also suffers a major limitation of the embrittlement effect which takes places due to cyclic absorption and desorption of hydrogen molecules. This repeated lattice expansion leads to a damage on the thin film surface, in terms of delamination, cracking, and blistering [41].

Several approaches have been considered by researchers to reduce this blistering effect by utilizing Pd nanostructures [42]. It has been reported by several researchers that the Pd nanostructure provides an improved sensing response compared to the Pd film, due to its high surface-to-volume ratio, which induces a large surface available for the interaction of Pd with surrounding hydrogen molecules [43–47]. Although the embrittlement was reduced at the nanoscale, the pure Pd has poor adhesion with the sensing element along with the hysteresis effect [48]. Development of Pd alloys' composition with other noble metals has also been proposed as an alternative strategy to suppress the phase transition of Pd and to detect a large range of gas concentration [49]. It was observed by several researchers that below 600 K, the isotherms of hydrogen absorption by Pd layer showed hysteresis effect related to first-order phase transition or, more specifically, to the separation of a diluted phase and hydride. According to the experiments, the addition of even a small amount of the second metal, e.g., silver (Ag), gold (Au) or tantalum (Ta), can apprecia-

bly suppress hysteresis. The Pd alloy with reduced hysteresis and embrittlement effect along with excellent adhesion has expanded its application in hydrogen sensing [20,29,50]. Alloying of Pd has already been studied with Au [51,52], Ag [53,54], yttrium (Y) [37,55], nickel (Ni) [56], and Ta [57], showing sensing response improvement, besides its structural stability, and repeatability. The optimization between the stability and the sensitivity has been performed by tuning the percentage composition of the alloy compounds. It has been reported that the hysteresis and the optical properties were enhanced for Pd-rich alloys [51]. The major challenges associated with hydrogen sensors have been attempted to be resolved by several detection techniques reported in the past years. Each detection technique, along with its sensing response, is highlighted in next section.

### 3.1. Evanescent Field (EF) Based Hydrogen Sensor (Side Polished/Tapered/Etched Standard Fiber)

Generally, an EF-based hydrogen sensor is the most common sensor available in the market. The sensing configuration comprises a tapered, etched, or side-polished optical fiber coated with the sensing layer, as shown in Figure 4. The tapering of optical fibers is mainly performed by a conventional fusion splicer [46,47], while the etched optical fiber configuration can be fabricated by hydrofluoric acid or laser etching techniques [58–60]. The side-polished optical fiber configurations are fabricated by precisely polishing the cladding region of conventional optical fibers [61,62]. The basic sensing mechanism of the EF-based hydrogen sensor depends on the interaction of evanescent field at the interface of the optical core fiber and sensing layer. These evanescent field decays exponentially corresponding to increasing the distance from the exposed core, and the attenuation coefficient depends on the refractive index of the fiber cladding [63]. These techniques were exploited to increase the intensity of the evanescent field at interface and improve the light-matter interaction to the hydrogen. The presence of hydrogen can be detected by monitoring the changes in the transmitted optical signal.

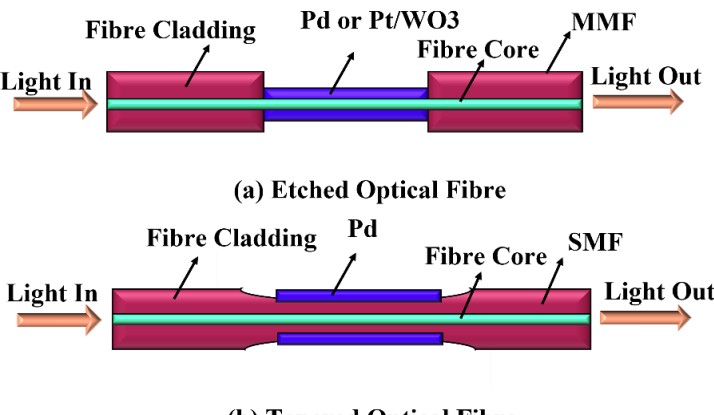

**Figure 4.** Schematic diagram of various sensing configurations based on evanescent field [64,65].

The first EF-based hydrogen sensor was reported by Azar et al. in 1999 [64]. The sensor was fabricated by depositing a 10 nm thin Pd layer on a 1.5 cm long etched section of a multi-mode fiber (MMF), as shown in Figure 4a. With the reported sensors, the authors detected hydrogen in the range of 0.2–0.6% with response times of 30–20 s at room temperature. In 2000, Sekimoto et al. reported [65] a similar configuration but coated with a $Pd/WO_3$ composite as shown in Figure 4b. The author reported a comparative study between two approaches: (a) using $Pd/WO_3$ composite layer containing silicone resin as the cladding; and (b) using a thin $Pt/WO_3$ cladding prepared by a sol-gel process. In presence of the hydrogen environment, a strong evanescent-wave absorption appears due to the formation of tungsten bronze. The result of the fabricated sensor with the $Pd/WO_3$ showed a very slow response, which was significantly improved by using $Pt/WO_3$. Similarly, in 2003, Villatoro et al. firstly performed hydrogen sensing using a tapered single mode fiber (SMF) coated with Pd film [66]. The sensing response of the sensor was adjusted by

tuning the taper length and its diameter. The maximum variation of the transmission was observed to be 50% when it was exposed to 4% of hydrogen. Later on, in 2007, Kim et al. fabricated another hydrogen sensor using side polished SMF coated with Pd film as shown in Figure 5 [67]. The transmitted optical power depends on the complex refractive index of the Pd and its thickness. When Pd interacts with hydrogen, its volume increases while the volume density of free electrons decreases. This leads to a decrease in both the real and imaginary parts of the refractive index of the Pd film. The sensing response of the device was tunable by adjusting the thickness of the Pd film. The fabricated sensor reported a response time of 100 s obtained from a 40 nm thick Pd layer.

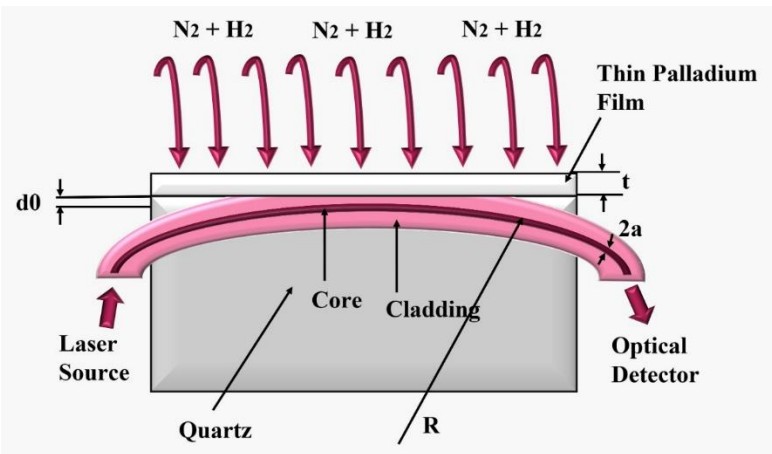

**Figure 5.** Schematic of a polished fiber-based hydrogen sensor [67].

In 2007, Moreno et al. reported an alternative way to excite the evanescent wave utilizing a hetero-core structure [68]. The sensing head comprised a small section of SMF (3, 5, 8 and 10 μm diameter) partially covered with a Pd/Au thin film spliced between two MMF as shown in Figure 6a,b. The core diameter mismatch between two fiber structures excites the cladding modes in SMF. When the sensor was exposed to the hydrogen environment, the refractive index of the sensing layer reduced resulting in the changes to the evanescent fields. With their proposed configuration, the authors reported an improved response time of 15 s by using Pd/Au under critically low hydrogen concentrations below 4%. Later on, in 2009, the same group improved the response time further to 4.5 s by utilizing a Pd/Au multilayer stack. The multilayer stack was composed of double layers of thin Pd (1.4 nm) and Au (0.6 nm) film deposited by the thermal evaporation method [69].

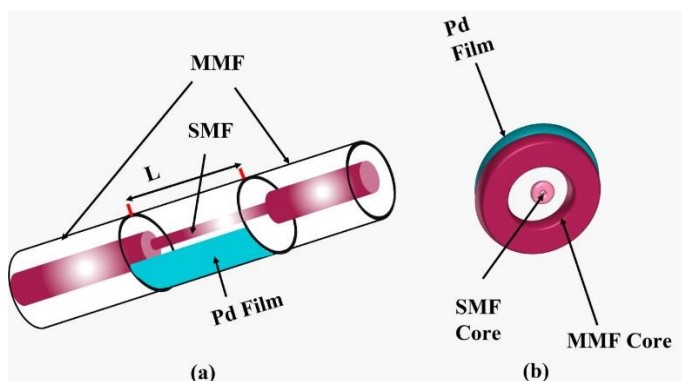

**Figure 6.** Schematic diagram of the hetero-core based hydrogen sensor [69].

In 2018, Li et al. demonstrated the sensor with the enhancement of sensitivity, stability, and response time by utilizing uniform Pd nanoparticles embedded in poly(methyl methacrylate) (PMMA) composites coated on microfiber as a sensing layer, as shown in

Figure 7 [70]. A ~20 μm thick sensing layer was coated on SMF of reduced diameter of ~57.93 μm. PMMA is a porous and highly permeable material for hydrogen molecules which helps to improve the sensitivity and the response time of the device. The amorphous PMMA and Pd nanoparticle composite turned Pd into palladium hydride (PdHx) in presence of hydrogen molecules, leading to the change in the effective refractive index and hence shifted the absorption peak wavelength. The sensor exhibited an average sensitivity of 5.58 nm/% along with an excellent response time of 5 s for 0.2–1% of hydrogen concentration. However, further optimization was required to reduce the agglomeration of Pd nanoparticles in the PMMA composites. In the reported experimental setup, the tungsten micron taper with diameter ~120 μm in its middle section was used to transfer one drop of the sensing material solution onto the surface of the silica microfiber which was placed on MgF$_2$ (Magnesium fluoride) substrates of refractive index 1.37 in order to reduce the loss of light. The Pd nanoparticles can absorb hydrogen molecules to form the PdHx species and change the effective refractive index of composite film, which produced the wavelength shift of resonance peak.

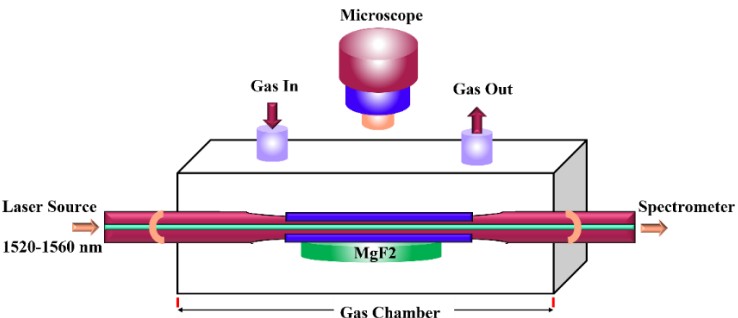

**Figure 7.** Microfiber based hydrogen sensor [70].

In 2021, Dai et al. reported a novel composition of WO$_3$-Pd$_2$Pt-Pt coated over a standard SMF, as shown in Figure 8 [71]. The sensor exhibited a rapid response to 100 ppm concentration of hydrogen at room temperature. The obtained results demonstrated an excellent resolution at 5 ppm for a broad range of concentration varying from 100 to 5000 ppm. As one of the most recent articles published in 2022, Alkhabet et al. reported a palladium nanoparticle coated on tapered optical fiber sensor to detect hydrogen at room temperature [72]. To enhance the evanescent field at sensing area, an MMF was tapered from 125 μm to 20 μm with 10 mm waist-length and 5 mm up and down tapered sections. The fabricated sensor operated at room temperature in the presence of 2% hydrogen concentration. The sensor exhibited a good sensitivity of 18.645% along with an average response and recovery time of 50 s and 230 s, respectively. In the same year, Kim et al. reported a new localized SPR based hydrogen sensor by capping Au nanoparticles with palladium [73]. The sensor was fabricated on a 3 cm length of MMF coated with Au capped Pd and its linearity was first validated with varying refractive index solutions. Following the result, the sensing configuration was implemented on various hydrogen concentrations varying from 0.8–4%. The reaction time at each concentration was observed to be 116 s until the signal was stabilized. The sensors reported a low LOD of 0.086%. Although the sensor showed a low LOD, it had a minor hysteresis for 1% of hydrogen.

In addition to these sensing configurations discussed above some other sensors are also reported in the past few years with some advantages and limitations. Table 2 illustrates the reported evanescent-based hydrogen sensors along with the sensitivity, detection range, and response time.

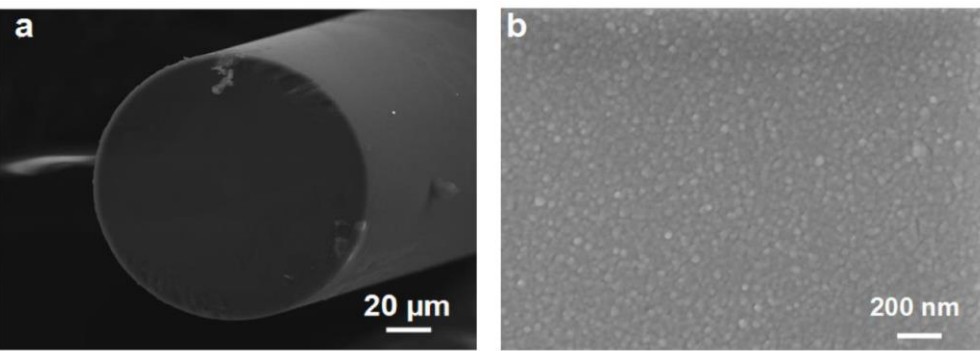

**Figure 8.** SEM image of (**a**) fiber tip (**b**) $WO_3$-$Pd_2Pt$-Pt composite film [71].

**Table 2.** Evanescent based optical hydrogen sensors using Pd and Pd-alloys.

| Sensing Material | Sensor | Detection Range | Sensitivity | Response Time | LOD | Reference |
|---|---|---|---|---|---|---|
| Pd film | Etched MMF | 0.2%<br>0.6% | NA | 30 s<br>20 s | N/A | [64] |
| Pd/$WO_3$ | Etched MMF | 100% | NA | 10–20 min | N/A | [65] |
| Pd film | Tapered SMF | 0–10% | NA | <100 s | 4% | [66] |
| Pd film | Polished SMF | 4% | NA | Response time 100 s/<br>recovery time 150 s | 4% | [67] |
| Pd/Au film | SMF sandwiched by 2 MMFs | <4% | NA | 15 s | N/A | [68] |
| Pd/Au multilayer stack | Hetero core | 4% | NA | Response 4.5 s/<br>recovery 13 s | 4% | [69] |
| Pd particle embedded in PMMA | Tapered SMF | 0.2–1% | 5.58 nm/% | 5 s | 35.8 ppm | [70] |
| Molybdenum Trioxide | Tapered MMF | 0.125–2.00% | 11.96 vol/% | 220 s | N/A | [74] |
| Pd particle | Tapered MMF | 2% | NA | Response 50 s/<br>Recovery 230 s | 2% | [72] |
| Pd | Tapered SMF | 1.8–10% | NA | <100 s | 2% | [75] |
| Pt/$WO_3$ | Etched MMF | 1% | NA | 5 min | N/A | [76] |
| Pd/Au | Etched D-shape fiber | 0.25–20% | NA | ≈30 s | N/A | [77] |
| Au capped with Pd | MMF | 0.8–4% | NA | 116 s | 0.086% | [63] |

### 3.2. Grating Based Hydrogen Sensor

The first optical fiber Bragg grating (FBG) based hydrogen sensor was developed by Sutapun et al. in 1999 [64] and following that the growth of the grating-based hydrogen sensor has attracted considerable attention thereafter. In the same year, Tang et al. developed a similar sensor to improve the sensing response by coating the sensor with Pd tubes of different thickness [78]. However, the response time of the device was recorded to be more than 200 min, which was the major limitation of that sensor. In the reported work, authors also investigated the effect of operating temperature on the device and observed a good wavelength shift for temperatures varying between 23–45 °C. In addition, the response time was significantly improved from 200 min to 2 min when operated at high temperature up to 95 °C. It was clearly identified that the sensor was highly influenced by temperature and showed a rapid response time at a higher temperature. Considering the response of the FBG hydrogen sensor in 2006, Trouillet et al. compared the performance of FBG and long period fiber grating (LPFG) coated with 50 nm Pd film [79]. The sensing response was examined in the presence of 4% hydrogen; the wavelength shifts of FBG and LPFG were approximately

14 pm and 7 nm (fundamental mode), respectively. The wavelength shift of the LPFG was observed nearly 500 times higher compared to the FBG. Kim et al. reported a series of experiment utilizing LPFG as the sensing platform [80,81]. Their study was reported in 2008 on a pair of LPFG based on a Mach-Zehnder interferometer. From the experimental results a fine structured interference fringe was observed in the transmission spectrum using a pair of LPFG, and therefore improved the resolution of the sensor compared to a single LPFG. The sensor exhibited a blue shift of 2.3 nm with a sensitivity of −0.29 nm/min for 4% of hydrogen concentration. However, until now the reported LPFG based hydrogen sensors utilized only Pd thin film, not Pd alloy. The LPFG based hydrogen sensors have not been exploited well due to their major limitation in their structure. In general, the LPFG-based hydrogen sensors are robust and easily reproducible, while the sensing head can perform only for the transmission type and hence difficult to be employed in remote sensing. Additionally, the multi-interferences between the core mode and various cladding modes make the transmission spectrum of LPFG very complicated, and therefore make it difficult to demodulate the spectrum signal.

Later in 2009, the sensing response was further improved by Buric et al. by utilizing double cladded fiber coated with sputtering glue (20 nm) and Pd film (200 nm). In the presence of hydrogen, the palladium expands and induces strain in the FBG leading to a shift in the Bragg wavelength. This wavelength shift can be accurately monitored and used to determine the concentration of hydrogen. In order to improve the sensitivity and response time of the device at a low hydrogen concentration range, the sensor was heated using an in-fiber laser [82]. This sensing configuration allows double interaction between the transmitted light and Pd film, which significantly enhances the sensitivity. The sensing response was repeatable with less hysteresis at room temperature and lower temperature of −50 °C. The sensor achieved a high sensitivity, fast response time of 10 s, and repeatability especially at low temperature. Even though the sensor exhibited a good sensing response, its operation was limited to a lower temperature which made it less effective for real-time monitoring. In order to improve the sensing response for real-time monitoring, an alternative approach was developed by the same group in 2009 [83]. In this case, the sensor was fabricated by inscribing 1 cm long gratings in high attenuation fiber (HAF) and later spliced with SMF as shown in Figure 9. The heating efficiency of the device was significantly improved by this technique allowing the sensor to detect 1% of hydrogen at −150 °C. Although the sensor was able to perform at 1% hydrogen, it was still suffering from the hysteresis effect for the Pd layer varying between 150–500 nm.

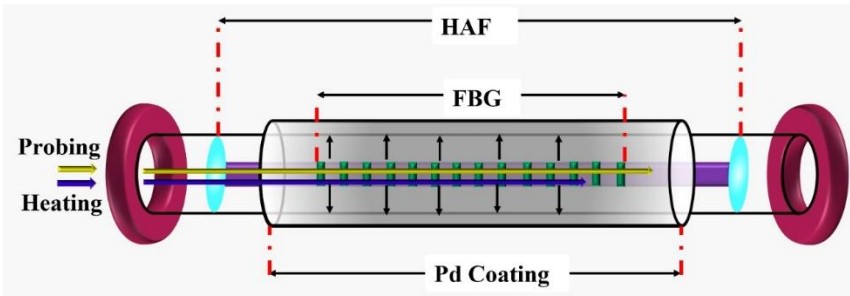

**Figure 9.** Schematic diagram of in-fiber hydrogen sensor with HAF [76].

In order to monitor a low hydrogen concentration below 1% another approach has been developed by reducing the diameter of the fiber up to tens of micrometers [84–86]. Reducing the diameter and modifying the FBG provides an enhancement in the evanescent wave and improves the sensitivity compared to standard or non-etched FBG, LPFG, and TFBG. In 2013, Silva et al. proposed a reported pair of FBG in one fiber to improve the sensitivity and compensate the effect of temperature simultaneously [87]. The FBG was written on a 50-μm diameter tapered single mode fiber by deep ultraviolet femtosecond laser technology while the second FBG was written on the 125 μm single mode fiber

section. The tapered portion was coated with 150 nm thick Pd film for hydrogen sensing while the other standard FBG was left uncoated for temperature compensation, as shown in Figure 10. The sensor exhibited a good sensitivity of 81.8 pm/% for hydrogen concentration varying between 0.1% and 1% (*v/v*) in the nitrogen environment. The sensor showed a good response time with a promising range but had poor adhesion.

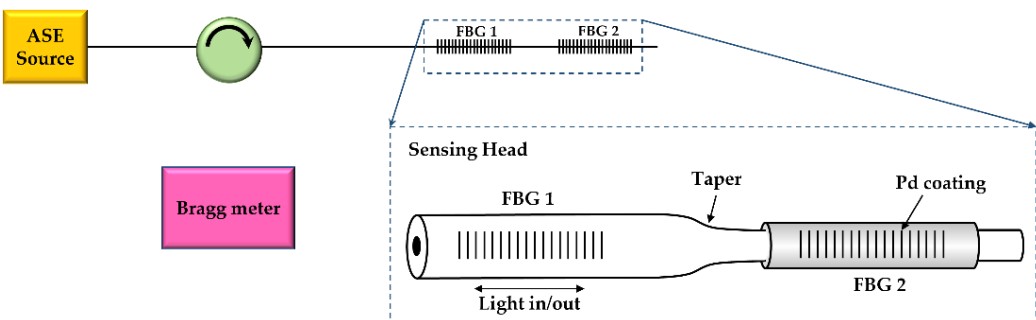

**Figure 10.** Schematic diagram of a tapered FBG based hydrogen sensor [87].

Although a pair of FBG significantly improved the sensing response and compensated the effect of temperature simultaneously, the adhesion of pure Pd film persisted as a challenging task for continuous monitoring. Some researchers considered using polymer coating as an intermediate adhesive layer between Pd and fiber to improve the adhesion [88,89]. This approach can significantly reduce the blistering of Pd film and improve the reproducibility. However, a thicker layer of polymer may decrease the light-matter interaction and hence affect the sensing response. Therefore, the optimization of the Pd alloy was extensively studied to improve the adhesion and sensing response simultaneously with negligible hysteresis [90]. In 2012, Dai et al. Reported the first Pd alloy composite coated on a FBG sensor [91]. The sensing head composed of an etched FBG of 17 μm diameter covered with a $Pd_{91}Ni_9$ composite. Authors also performed X-ray diffraction (XRD) after the experiment to illustrate a good structural stability in the presence of the hydrogen. The sensor exhibited nearly a 15 pm spectral shift towards the 1% hydrogen concentration at room temperature. The sensor showed a good spectral shift but a slow response time of 5 min which limits its application. This was later improved by utilizing a Pd:Ag = 76:24 composite on a similar sensing configuration [53]. Although the sensor exhibited good repeatability, the sensitivity of 10 pm/% was obtained. The sensitivity was later enhanced by the same group in 2014 in which a polypropylene sheet was used as a flexible substrate [56]. The sensor illustrated an excellent spectral shift of 146 pm towards 4% hydrogen, along with a good linear response, and repeatability.

In some research for the Pd/Ag, the sensor exhibited a lower sensitivity but a faster response time when operating at temperature varied from 20 °C to 80 °C [92]. A comparative investigation was reported by Samsudin et al. in 2016 [93]. This article reported two sensing configurations comprising of Pd and chromium (Cr) composites in ratio of 100:0 and 58:42, respectively. The adhesion of the sensing layer was improved by intermediate $TiO_2$ layer. The reported sensor showed 15 pm spectral shift with the hydrogen concentration varied from 0 to 650 ppm. The sensor exhibited a low sensitivity because the standard FBG was used instead of FBG on a reduced diameter fiber. The study showed that the Pd and Cr compositions can detect a higher concentration range of hydrogen compared to the pure Pd film. A side polished FBG is considered to be an intrinsically sensitive to curvature, which was utilized by several researchers to enhance the sensitivity of hydrogen sensor [94,95]. A side polished FBG is usually made by polishing the surface by motor-driven polishing wheel [96–98]. The sensing response of the sensor was observed to have increased by 100% compared to the standard FBG [99]. Dai et al. reported the first side polished FBG based hydrogen sensor using a $Pd/WO_3$ composite [99]. The sensor exhibited a maximum wavelength shift of 25 and 55 pm for 4% and 8% of hydrogen concentrations, respectively. Another similar configuration was reported by Luo et al. in 2016 to determine the dissolved

hydrogen concentration in power transformer oil [100]. The polished FBG was observed to be sensitive to the curvature due to stress arising from the sensing layer leading to a shift in Bragg wavelength. The sensor exhibited a high sensitivity of 1.96 µL/L. In 2018, Fisser et al. conducted a series of comparative experiment over Pd alloy coated FBG sensor [101–103]. One sensor comprised of 1600 nm thick Pd layer, and the other was coated with 20 µm or 100 µm thick Pd foil around FBG. The sensor exhibited a good spectral shift of 480 pm and 225 pm for 20 and 100 µm of Pd foil at 5% hydrogen concentration. However, 10 pm, 160 pm, and 90 pm, was observed for 1600 nm thick Pd layer, 20 and 100 µm thick Pd foil, respectively, towards 1% $H_2/N_2$ at 90 °C [102]. Although the sensitivity of the device was improved, all the sensors had a poor response time of several hours. In addition of the standard FBG, other grating based fibers were also utilized for the hydrogen sensing with improved sensing response [98]. In the same year 2018, Yu et al. demonstrated the titled FBG (TFBG) based hydrogen sensor coated with Pd membrane by the chemical coating method, as shown in Figure 11 [104]. The Pd metal was synthesized in aqueous solutions, which was later utilized to determine variation of the surrounding refractive index corresponding to hydrogen concentrations. The change in the refractive index leads to an unsynchronized variation in cladding modes and Bragg wavelength in the TFBG transmission spectrum. The cross-sensitivity of expansion strain and temperature variation, which was one of the major advantages of this device, was eliminated in the sensor configuration. However, the slow response time of 5 min limited its application in real-time monitoring. Recently, Shen et al. reported a novel TFBG based hydrogen sensor utilizing PdAu thin nanofilm over the fiber surface [105]. The sensor was fabricated by cascading an enlarged taper upstream to the TFBG. The cladding modes and Ghost mode of the TFBG were coupled into the core to form a reflective-type TFBG-based hydrogen sensor. The sensor exhibited a high sensitivity of 4.83 dB/% with a low LOD of 0.07% and response time of 26 s for the hydrogen concentrations varying from 0% to 0.7%.

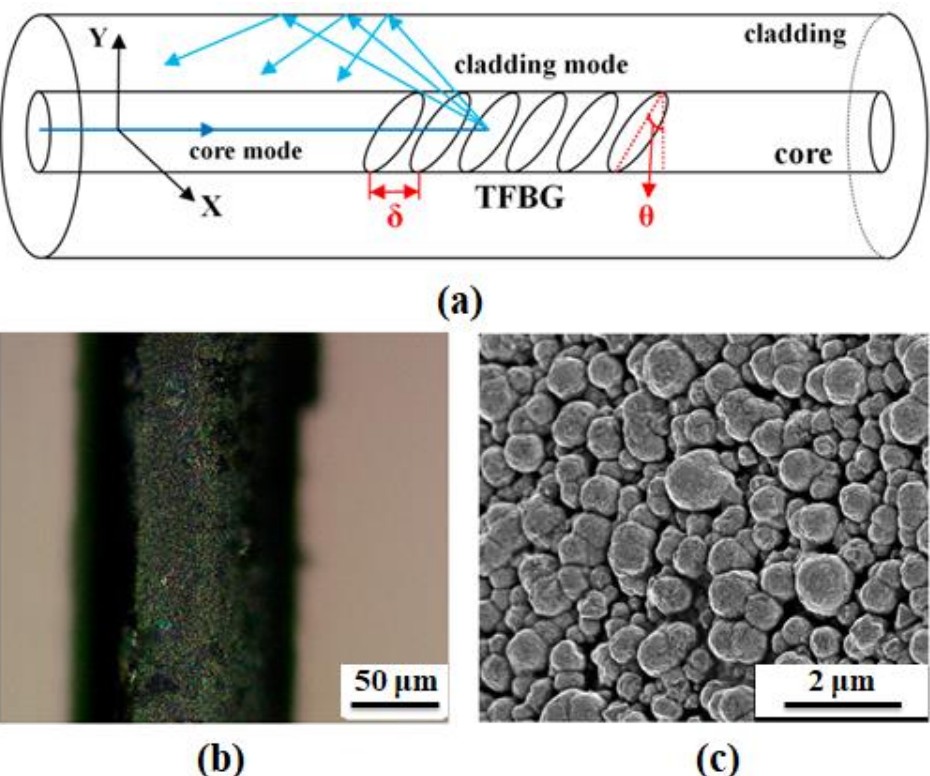

**Figure 11.** (**a**) schematic of TFBG (**b**) microscopic images of coated Pd layer on the Pd-plated fiber surface (**c**) SEM image of uniform Pd particles [104].

In addition to these FBG based hydrogen sensors discussed above there are also some more FBG based of sensors which have been reported in the past years with some advantages and limitations. Table 3 shows previously reported key FBG based hydrogen sensors along with their sensitivity, detection range, and response time.

**Table 3.** FBG based optical hydrogen sensor.

| Sensing Material | Sensor | Detection Range | Sensitivity | Response Time | LOD | Reference |
|---|---|---|---|---|---|---|
| Pd:Ag = 76:24 | Etched FBG | 4% in volume | 10 pm/% | 280–300 s | 4% | [53] |
| Pd:Ni = 91:9 | Etched FBG | 0.5–4% | 36.5 pm/% | 5–6 min | N/A | [56] |
| Pd film | LPFG | 4% | NA | NA | NA | [69] |
| Pd film | LPFG | 4% | −0.29 nm/min | NA | NA | [70] |
| Pd film | FBG/w HAF | 1–10% | 27 pm/% | - | N/A | [83] |
| Pd | Tapered FBG | 0.1–1% ($v/v$) | 81.8 pm/% | 30 s | N/A | [87] |
| Pd/Ti/polyimide | FBG | 0.25–2% | 13.5 ppm/pm | ≈1 h | N/A | [88] |
| Pd/Ti/polyimide | FBG | 1791.46 ppm | 0.042–0.044 pm/ppm | - | N/A | [89] |
| $Pd_{91}Ni_9$ | Etched FBG | 1% | 15 pm/% | 5 min | N/A | [91] |
| Pd/Ag | FBG | 0–2000 μL/L | 0.055 pm/(μL/L) | 24 min | 18 μL/L | [92] |
| Pd:Cr = 58:42 (with $TiO_2$) | Standard FBG | 0–650 ppm | NA | 10 min | 4% | [93] |
| $Pd/WO_3$ | Polished FBG | 0–8% | 6.5 pm/% | 40–90 s | N/A | [99] |
| $Pd/WO_3$ | Polished FBG | 0.2–1.4% | 196 μL/L | - | N/A | [100] |
| Pd | FBG | 1%/5% | NA | Response 20–30 min/ recovery 50 min | 1% | [102] |
| Pd foil | Etched FBG | 1–5% | 212.6 pm/% | 4 h | 1% | [103] |
| Pd membrane | Titled FBG | 1–4% | NA | 5 min | 1% | [104] |
| Pd film | FBG | 4% | NA | 2 min | N/A | [106] |
| $Pt/WO_3$ | FBG | 1500–20000 ppm | NA | 55–80 s | N/A | [107] |
| Pd/Ag | Side-polished FBG | 0.08% | 4770 pm/% | <1 h | N/A | [108] |
| Pd | Two Etched FGBs | 1% | 20 pm/% | 2 min | 1% | [109] |
| $Pd_{75}Ag_{25}$/Ni | FBG/w microgroove | 0–4% | 16.5 pm/% | 10 min | N/A | [110] |
| Pd | Tapered FBG | 5% | 216 pm/% | 1 min | N/A | [111] |
| $Pd_{91}Ni_9$ | FBG | 0–1% | 0.01 pm/ppm | ≈200 s | 4% | [112] |
| $Pt/MoO_3$ | FBG | 1500–15000 ppm | 0.022–0.031 pm/ppm | Response 100 s/ recovery 110 s | N/A | [113] |

### 3.3. Microstructured Optical Fiber (MOF) Hydrogen Sensor

In 2013, Zhou et al. demonstrated a microstructured optical fiber (MOF) in-line interferometer for hydrogen detection [114]. The end face of the MOF along with the outer surface was coated with a thin palladium layer whereas the other end was spliced to a SMF-28, as shown in Figure 12. The whole configuration formed a set of reflection-type all-fiber for hydrogen detection. The sensor was investigated over a wide range of hydrogen concentration varying from 0% to 5% and achieved a wavelength-shift of 1.2 nm when hydrogen concentration reaches 5%. Therefore, a compact, highly stable, and cost-effective sensor was achieved.

In 2015, Yang et al. utilized $Pd_{91}Ag_9$ alloy to develop a temperature-insensitive hydrogen sensor [115]. The sensing configuration comprised a photonic crystal fiber (PCF) coated with a thin Pd/Ag composite film. The birefringence of PCF was modulated by hydrogen absorption which induced deformation to the sensing layer, resulting in a spectral shift of the interference spectrum at the output of the Sagnac interferometer. The sensor

was investigated over 1–4% hydrogen concentrations. They evaluated two methods to improve the sensing response including an increase of (a) the Pd amount in the composite film, and (b) increase of PCF length. The result exhibited that the increasing amount of Pd in composite film enhances the sensitivity of the devices. The sensitivity was also observed to be improved by using a longer length of Pd/Ag coated PCF. It was observed that with a ~100 mm longer sensing element a good spectral shift of ~1.310 nm was achieved for hydrogen concentration varying from 0 to 1% and the sensitivity coefficient of ~131 pm% was achieved in the range of 1% to 4%. In 2018, Zhang et al. reported on the PCF based sensors [116,117]. It was observed that the reflective hydrogen sensor based on a fiber ring laser integrated with the PCF modal interferometer exhibited an excellent sensing response and high sensitivity [117]. The sensing head was fabricated by forming two collapsed regions on both ends of PCF which were later coated by Pd/WO$_3$ alloy. The major advantage of the fiber ring laser integrated with the PCF was the detection limit improvement allowing a high signal-to-noise ratio (SNR). Experimental results exhibited a good sensitivity of 1.28 nm/% compared to the previous configuration [116], along with high SNR of ~30 dB, and low detection limit of 0.0133%. However, the response time of the device was not reported.

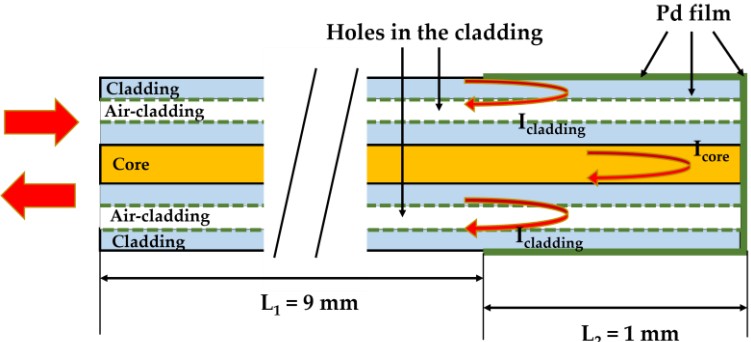

**Figure 12.** Schematic of MOF based hydrogen sensor coated with Pd film [114].

In order to develop a hydrogen sensor for continuous monitoring and for a long period of time, Aazi et al. reported a novel polarization maintaining Panda fiber embedded with Pd nanoparticles [118]. A comparative investigation between two prototypes with and without Pd embedded was performed. The reported sensor was based on the monitoring of the birefringence modulated by Pd nanoparticles. The integration of Pd particles with polarization maintaining-Panda fiber improved the sensing response by 26% and reduced the response time by 50%. It offers a promising way to improve the stability and long-term robustness of the fiber sensors. Although the fabricated sensor successfully resolved the instability issues, high sensitivity and fast response time were not achieved, which limit its application that requires real time detection. Overall, the poor sensing response of the MOF based hydrogen sensor limits its commercialization. Its complex structure also restricts its practical use in real world detection. To improve the sensitivity, Liu et al. recently reported a hollow fiber coated with ethylene-vinyl acetate (EVA)/Pd composition [119]. The fabricated device was comprised of EVA/Pd layers sequentially deposited on the inner surface of the silica capillary tube as shown in Figure 13. The theoretical ray transmission model was utilized to determine the influence of the thickness of EVA and Pd layers. The experimental results exhibited a red spectral shift with the increase of the hydrogen concentrations varying from 0–4%. An improved sensitivity of 2.66 nm/% was achieved in the range of 0–4% hydrogen concentrations. However, the response time which is one of the major limitations of this device was not studied.

In addition to the discussed MOF based hydrogen sensors, there are also some other similar sensors which have been developed in the past years and some of the key sensors are listed in Table 4.

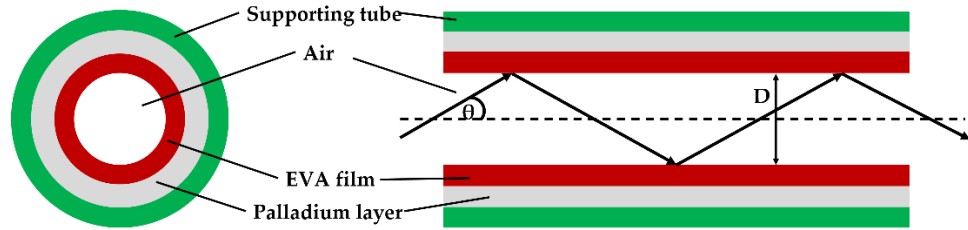

**Figure 13.** Schematic of hollow fiber-based hydrogen sensor.

**Table 4.** MOF based optical hydrogen sensors.

| Sensing Material | Sensor | Detection Range | Sensitivity | Response Time | LOD | Reference |
|---|---|---|---|---|---|---|
| Pd film | SMF-PCF | 0–5% | NA | - | N/A | [114] |
| Pd/Ag | PM-PCF | 1–4% | 131 pm/% | - | N/A | [115] |
| Pd/WO$_3$ | Fiber loop mirror-PCF | 0–1% | 1.12 nm/% | - | 0.14% | [116] |
| Pd/WO$_3$ | Ring laser-PCF | 0–1% | 1.28 nm/% | - | 0.0133% | [117] |
| Pd particle embedded | PM-Panda | 100% | NA | 20 h | N/A | [118] |
| EVA/Pd | Hollow core fiber (HCF) | 0–4% | 2.66 nm/% | - | N/A | [119] |
| Au/Pd/WO$_3$ | PCF | 0–3% | 0.19 nm/% | - | N/A | [120] |
| Pd/WO$_3$ | PCF | 0–10,000 μL/L | 0.109 pm/(μL/L) | 33 min | N/A | [121] |
| Pd-Pt | Hollow core-PCF | 0–100% | NA | 46 s | N/A | [122] |
| Pd | Taper-PCF | 0–6% | NA | - | N/A | [123] |
| Pd-Au-Graphene | Capillary-HCF | 0–1000 ppm | NA | 120 s | 741 ppb | [124] |
| NA | Tellurite PCF | 0–3% | −0.236 nm/% | - | 2500 ppm | [125] |

### 3.4. Plasmonic Fiber Hydrogen Sensor

More recently, the potential of plasmonic for sensitivity enhancement of hydrogen sensors has been exploited [126–128]. Metal and metal-alloy play key roles in the plasmonic nature of the optical field [129]. In 2002, Bévenot et al. reported first surface plasmon resonance (SPR) based sensing configuration for detection of hydrogen leakage [130]. The sensing configuration consisted of a thin Pd layer coated over a small section of uncladded MMF. The variation in a complex dielectric constant of Pd with the presence of hydrogen leads to the change in the SPR absorption spectra resulting in the loss reduction of the transmitted light at the Pd reflection interface. This effect was further improved by the selective launching of higher order modes via a collimated beam. From the reported sensor, a low hydrogen concentration of 0.8% in pure nitrogen was successfully detected along with a good response time of 3 s (pure hydrogen) and 300 s for the lowest concentrations. Later in 2011, Perrotton et al. reported a numerical investigation over a SPR based hydrogen sensor using wavelength modulation technique [36]. The reported approach required the deposition of a transducer layer, a combination of a multilayer stacking of silver, silica, and Pd on a small section of uncladded MMF. The sensor comprised a 100 nm thick silica to tune the resonant wavelength, while silver and Pd layers were of 35 nm and 3 nm thick, respectively, in order to enhance the detection accuracy and sensitivity. The reported configuration achieved a 17.6 nm shift in the resonance wavelength with the presence of 4% hydrogen. The main advantage of the device is its sensitivity towards intensity fluctuations. In this numerical investigation, the response time was not studied. In 2013, the same group validated a similar configuration experimentally, but with Au as the plasmonic layer [131]. The sensor consists of a multilayer of Au/SiO$_2$/Pd of thickness 35/180/3.75 nm, respectively, coated on a MMF. The sensitivity

and selectivity of the device was optimized for a 3.75 nm thin Pd layer. The fabricated sensor exhibited good sensitivity to a hydrogen concentration varying from 0.5 to 4% H2 in Ar, along with a good response time of 15 s. Later on, a series of experiment were performed by Hosoki et al. from 2013–2016 on SPR based hydrogen sensors using $Au/Ta_2O_5/Pd$ at different compositions [132–134]. The sensor coated with an annealed $Au/Ta_2O_5/Pd$ layer achieved an excellent response time [134]. The schematic of the sensing configuration is shown in Figure 14. The response time of the sensor was investigated by annealing the sensing layer at a temperature of 400 °C, 600 °C, and 800 °C. After annealing, nano-sized structural cracks appeared throughout the sensing layer which was responsible for sensing response enhancement from a larger surface area. After the hydrogen immersing process, the response time of the sensor was observed to be 8 s, which was significantly improved by two times more than the previous identical configuration (but without annealing) reported by the same group [132,133].

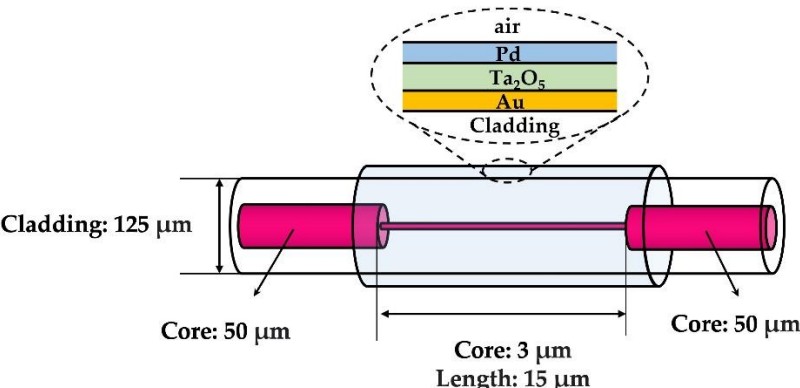

**Figure 14.** Schematic of multilayer SPR based hydrogen sensor based on hetero core fiber [134].

In 2016, Downes and Taylor reported a numerical investigation of a plasmonic hydrogen sensor using a multilayer capped with an alloy of Pd and Y as the hydrogen sensitive layer [37]. The sensor employs a multilayer stack of $Ag/SiO_2/PdY$ of thickness 50.5/72/3 nm, respectively, over the core of a MMF. The PdY alloy was considered to improve the lifetime of the device by reducing the mechanical stress on the sensing layer occurs during sensing operation. In the reported work, the author numerically investigated the performance of the sensor in terms of the sensitivity and detection accuracy for 4% hydrogen concentration. This sensor exhibited an average sensitivity and a detection accuracy of 17.64 nm and 0.014 nm$^{-1}$, respectively, towards the 4% hydrogen concentration. The sensitivity of the device was defined as the shift in resonant wavelength with respect to the overall change in surrounding gaseous hydrogen concentration. The Pd and Au alloy has also been reported as an excellent alloy composition to improve the sensing performance [135]. It was observed that increasing the amount of Au in the alloy nanoparticles up to 25 atoms % can suppress the hysteresis and embrittlement effect during absorption and desorption of the hydrogen and can increase the sensor accuracy to below 5% throughout the investigated range varying from 1 mbar to 1 bar hydrogen pressure. The dimensions, geometry, and substance of the nanoparticles, as well as its surroundings, or more particularly, its dielectric characteristics, all affect the plasmonic wavelengths at which the LSPR occurs [136]. Consequently, it is essential to identify modifications to the nanoparticle directly (through modification in dimension, structure, materials, or electrical characteristics) or to its local surrounds by monitoring the LSPR wavelength. Hydrogen could very well be detected via plasmonics in two distinct ways: directly and indirectly. Hydrogen detectors made of hydrogen-forming metallic nanostructures, including Pd, are used in the direct technique. Thus, the term "direct hydrogen sensing" refers to the utilization of nanoscale particles' LSPR to monitor hydrogen absorption in the metals. Hydrogen leads to the swelling of metallic structures, which results in the variation of the metal's permittivity [137]. These influences will have an impact on the LSPR's oscillation frequencies that are employed as the output response of the device. Langhammer

et al. published an impressive perspective on nanostructured Pd-based hydrogen sensors which emphasized their limitation for commercialization [138–141]. One key application of the large-area nanostructured hydrogen sensor was first demonstrated by Langhammer et al. who utilized palladium nanodisks to perform hydrogen sensing and study the hydrogen-induced phase transition from Pd to PdH on the nanoscale [138]. The transition leading both the expansion of the Pd lattice and the strong change of the dielectric function, allowing it to be observed in resonant plasmonic systems which was later utilized to determine hydrogen concentration. For the Pd nanodisk sizes under consideration, a linear correlation was discovered and subsequently mathematically confirmed relying on first simulations by Poyli et al. [142]. Additional nanostructures, such as Pd nanorings [143] and concave Pd nanocubes, have also been designed for direct LSPR hydrogen detection in addition to Pd nanodisks [144]. For these kinds of detectors to be capable of competing with some other kinds of hydrogen detectors in practical implementations, further research in this regard is essential. Different nanomaterials have lately been researched into for use in direct LSPR sensors. According to Sil et al. [145] substrates coated with a gold hemisphere between 20 and 36 nm in diameter have a plasmonic response when hydrogen is introduced. Mukherjee et al. published the very initial research on this type of plasmon-induced hydrogen ionization on nanostructures in 2012 [146]. Another study by Strohfeldt et al. [147] showed that it is feasible to move the LSPR on and off by exposing yttrium (Y) nanostructures to hydrogen. In particular, the first exposure of the yttrium particles to hydrogen results in the formation of the dielectric yttrium trihydride ($YH_3$), which effectively suppresses the LSPR. The authors assert that a range of unique hydrogen enabled a plasmon switching techniques may be achieved with the aid of yttrium nanoparticles as a fundamental structural component. Devices with switchable perfect absorbers or plasmonic electromagnetically induced transparencies are examples of potential uses.

When a nanoentity interacts with hydrogen, its LSPR (or optical cross-sections, to use the more generic terminology) can be extremely feeble to be useful for hydrogen detection in particular circumstances. In these circumstances, it is conceivable to use additional nanostructure with better and more specialised LSPR capabilities as detectors that examine the hydride-forming objects nearby (a couple to just several tens of nanometer range away) via their strong optical near fields. In this situation, the LSPR nanostructures, which are normally made of gold, should ideally not interact with the hydrogen, and instead solely serve as antenna and signal transduction. This detecting method is known as indirect detection. The earliest known indirect LSPR hydrogen sensors were focused on different composites made of oxides and, generally, Au nanostructures rather than using metal hydrides as the reactive ingredient. For this kind of sensor, the dissociated hydrogen adsorption, permittivity changes in the oxide, or movement of electrons from the oxide to the plasmonic Au nanoparticles all contribute to the high point shifting output [148]. Hence, it is an advantageous if the hydrogenation observations are conducted at the solitary nano structures level in order to comprehensively bypass such restricting effects. Additionally, single-particle tests would offer researchers the opportunity to thoroughly define the nanoparticle under study, enabling them to tie any impacts on hydrogenation performance to the particle's dimensions, form, and most prevalent surface aspects. It appears that the indirect LSPR-sensing approach, which entails positioning a plasmonic antenna next to a hydride-forming entity, enables the execution of single-particle hydrogen-sensing studies. In essence, the experiment is rather simple because all that is required to illuminate and gather the scattered light from the desired nanoparticles is an optical dark-field microscopy. Liu et al. conducted the very initial significant indirect hydrogen-sensing research on a singular Pd nanoparticle [149] Shegai et al. developed a similar strategy utilising a different sample setup [150]. The hydride former (Pd or Mg) was placed on top of a stack of Au nanostructures, followed by an insulating spacer. This has firmly shown that it is actually feasible to analyse hydrogen absorption in individual nano-entities. The hydride-forming Pd and Mg nanoparticles under investigation were, however, quite big (30–60 nm) in both aforementioned situations. Previous research on nanoparticle

assemblages has demonstrated that the intriguing domain where one may anticipate impacts enforced by the particle diameter to emerge perceptible is below 10 nm [151–153]. In 2018, Augroho et al. utilized the feature of PdAu alloy composition and reported a nano plasmonic hydrogen sensor decorated with nanostructured using colloidal lithography and electron beam lithography [154]. The reported sensor was developed by depositing the arrays of nanofabricated alloy and metal nanostructures over optical fiber through pattern transfer. The nano plasmonic hydrogen-sensor made of nanofabricated arrays of Au, Pd and $Pd_{75}Au_{25}$ alloy nanoparticles onto a flat support was transferred onto the unclad tip of optical fiber with 300 μm diameter of $SiO_2$ core and hard fluoropolymer cladding. The hysteresis effect was successfully supressed, and additionally improvement of the response time of <5 s was achieved.

Since 2018, the Pd/Au alloy composition has attracted huge attention due to its ability to eliminate the hysteresis effect, enhance the sensitivity, and improve the response time [127,141,155,156]. Cao et al. reported a D-shaped fiber coated with PdAu alloy nanostructures as shown in Figure 15 [77]. The sensor was fabricated by depositing Pd/Au alloy over $SiO_2$ to form nanocone structures which was later transferred to the polished D-shaped fiber by using a maskless ion etching method. The mutual interaction between EF and the nanostructures can improve the sensitivity for hydrogen detection varying from 0.25% to 10% (by volume) in air. The sensor exhibited a hysteresis free effect along with a rapid response. Recent study published in 2022, Kim et al. reported a localized surface plasmon resonance (LSPR) based hydrogen sensor [73]. The sensing configuration consisted of palladium-capping on a 3 cm length of uncladded MMF. The sensor was validated with refractive index solutions to confirm the linearity response before undergoing the hydrogen test. Later, the sensor was tested with 0.8% to 4% concentration of hydrogen. The response and recovery times for 4% concentration were observed to be 107 s and 126 s, respectively.

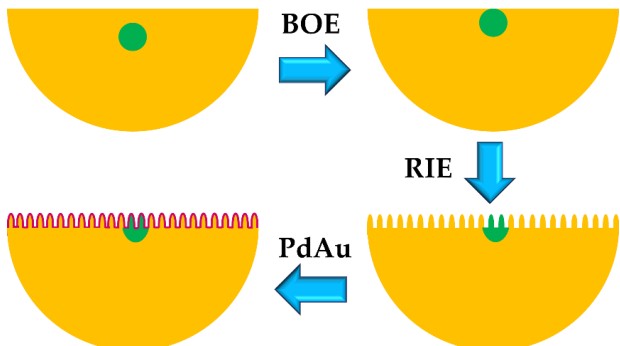

**Figure 15.** Schematic of D shaped fiber coated with PdAu alloy.

In addition of the above sensors, there are some more SPR based sensors which have been developed in the past years, which are listed in Table 5.

**Table 5.** Plasmonic optical hydrogen sensor.

| Sensing Material | Platform | Detection Range | Sensitivity | Response Time | LOD | Reference |
|---|---|---|---|---|---|---|
| Au-IRMOF-20 | MMF | 0–50% | NA | 5 s–10 s | N/A | [10] |
| $Ag/SiO_2/Pd$ | MMF | 4% | NA | - | N/A | [36] |
| $Ag/SiO_2/PdY$ | MMF | 4% | NA | NA | NA | [37] |
| $Ag/ZnO_{(1-x)} Pd_x$ | MMF | 4% | NA | 1 min | N/A | [128] |
| Au/silica/Pd | MMF | 0.5–4% | NA | 15 s | 0.5% | [131] |
| $Au/Ta_2O_5/Pd$ | Hetero core | 4% | NA | 15 s | N/A | [132] |

**Table 5.** *Cont.*

| Sensing Material | Platform | Detection Range | Sensitivity | Response Time | LOD | Reference |
|---|---|---|---|---|---|---|
| $Au/Ta_2O_5/Pd$ | MMF | 4% | NA | 25 s | N/A | [133] |
| $Pd_{75}Au_{25}$ | MMF | 0–4% | NA | Response time 90 s/ Recovery time 10 s | N/A | [154] |
| $Au/Ta_2O_5/Pd$ | Hetero-core fiber | 0–4% | NA | 40 s | N/A | [157] |
| $Ag/Si/Pd$ | Plastic clad MMF | 4% | NA | - | N/A | [158] |
| $Ag/Si/WO_3/Pt$ | MMF | 2% | NA | - | N/A | [159] |
| Au-Pd nano cube | Fiber bundle | 4% | NA | Response time 30 s/ recovery time 4 s | N/A | [160] |
| $Ag/SiO_2/Pd$ | MMF | 0–4% | NA | - | N/A | [161] |
| $Ag/TiO_2$ | MMF | 14.7% | 523 nW/% | - | N/A | [162] |
| Graphene-Au-Pd nanofilm | FBG | 0–4.5%, | 290 pm/% | - | N/A | [163] |

## 4. Other Approaches for Hydrogen Sensing

In addition to these classic optical fiber-based hydrogen sensors, some novel techniques were developed and exploited widely in the past few years including in SAW, stress based, microcantilever, micromirror, etc. These are briefly summarized here.

### 4.1. SAW Based Hydrogen Sensor

Surface acoustic wave (SAW) gas sensors generally determine the variation in the properties of acoustic waves; they arise due to an adsorbate on the surface or absorbate on the piezoelectric material. The first SAW based gas sensor was reported by King in 1964 based on the monitoring of bulk acoustic waves in a piezoelectric quartz crystal [164]. In the case of hydrogen, the Pd hydride structure undergoes a phase change when hydrogen absorption increases, resulting in the formation of the β-phase, which breaks down the crystal structure and leads to the variation in stiffness, density, and elasticity [24]. In 1995, Anisimkin et al. reported a SAW supported hydrogen sensor to monitor the variation in density and elasticity of Pd in presence of hydrogen [165]. SAWs are sound waves which propagate parallel along the surface of an elastic material, with their displacement amplitude decaying into the material so that they are confined to within roughly one wavelength of the surface [166–168]. The basic sensing mechanism of the SAW based sensor is attributed to the fact that most of the acoustic energy is localized near the surface of the device within one or two wavelengths [169]. Therefore, any significant change in the chemical or physical properties of the sensing material, such as electric loading, mass loading, elastic loading, temperature, pressure, and density, will have a significant impact on the propagated acoustic waves [165]. For hydrogen sensing, SAW is generated by utilizing piezoelectric substrate as a sensing platform comprised of two sets of interdigital transducers coated onto it. Among these two transducers one is used to convert the input electrical signal into the acoustic wave, while the other converts it back to the electrical output signal. Generally, $LiNbO_3$ is considered as the most popular piezoelectric substrate to produce SAW [169–172]. However, one researcher also used $LiTaO_3$ [173]. The first SAW supported hydrogen sensor was reported by D'Amico et al. in 1982 [174]. Since then, the techniques have been exploited by several researchers in order to improve sensing response of hydrogen sensor [7]. In 2019, Li et al. reported a hydrogen sensor based on a AlN/Si layered structure coated with Pd nanoparticles decorated over graphene oxide [175]. Aluminum nitride films have received huge attention due to their high acoustic wave velocity, good chemical and temperature stability, and CMOS process compatibility [176,177]. The sensor exhibited a ~9.5 times higher sensitivity compared to the other sensors based on only

graphene oxide along with good stability, low cross-sensitivity, and a fast response time of 7 s. In the same year of 2019, Wang et al. reported an improved response time using Pd/Cu nanowire deposited on LiNbO$_3$ substrate [178]. The adsorption of hydrogen molecules in Pd/Cu nanowires modulates the propagation of SAW by the induced acosuto-electric coupling effect arising from the changes in the resistivity of Pd, the shifts in oscillation frequency corresponding to various gas concentration was monitored as the sensor signal. The larger surface-to-volume of the Pd/Cu nanowire helps in the improvement of its sensing response. The fabricated sensor was investigated using the differential oscillation loop and exhibits excellent response time of 4 s with good repeatability and high sensitivity of 1.5 kHz/%. Another SAW based hydrogen sensor was recently reported by Devkota et al. using indium oxide (IO) and indium tin oxide (ITO) [179]. Authors investigated the sensing potential of IO doped ITO deposited over langasite based acoustic wave reflective delay line sensor to monitor hydrogen at a higher temperature of 350 °C. The sensor exhibited a maximum sensitivity of 0.0018 rad/vol% for the concentration varying from 0 to 100%. Hydrogen has a high sound velocity of 1314 m/s compared to that of air, 346 m/s at 298 K. Some researchers have also exploited this feature in the development of hydrogen gas sensors [180,181]. Although the reported sensor performed effectively, the response time was not up to the mark to realize it for commercialization. In order to overcome the challenge, Wang et al. reported a Pd/Ni nanowire coated SAW hydrogen sensor, as shown in Figure 16 [182]. The sensor was fabricated by depositing a Pd/Ni nanowire on a SAW device and the gas adsorption in the Pd/Ni nanowire modulates the conductivity, and perturbs the SAW velocity, and the corresponding variation in phase was collected as the sensing signal. The sensor exhibited a high sensitivity of 1.65 mV/% with extremely fast response time of less than 2 s and a low LOD of 7 ppm. The velocity of sound in Pd has also been exploited in order to monitor the hydrogen leakage. The acoustic wave velocity in a solid waveguide depends on the material density and on the amount of stress and strain to which it is subjected [7]. When Pd interacts with hydrogen it modulates its density and mechanical properties which can therefore be detected as a difference in the velocity of sound waves propagating through it. SAW based hydrogen sensors have the potential to detect hydrogen over a wide range of concentration varying from ppm level to 100%, along with a rapid response time. However, major improvements are required in terms of its long-term stability and their sensitivity to temperature and selectivity to hydrogen in mix gas environment.

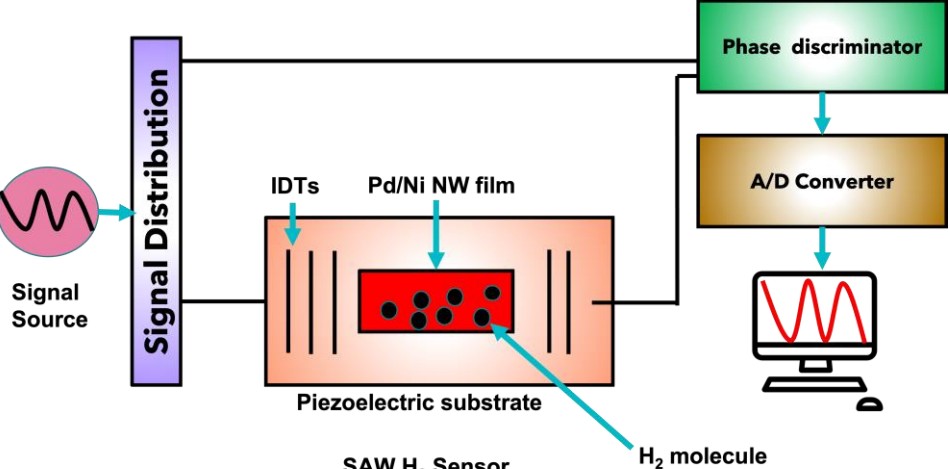

**Figure 16.** Pd/Ni nanowire coated SAW hydrogen sensor.

*4.2. Hi-Birefringence (Hi-Bi) Based Hydrogen Sensor*

In the presence of hydrogen, Pd absorb hydrogen molecules and breaks them into atoms, which subsequently induces stress due to the lattice expansion of the material [165]. There are several optical fiber-based sensors which have been developed to monitor the stress variation in Pd. The first stress monitoring was reported by Sutapun et al. in

1999 by utilising Pd thin film on the surface [64]. The sensing mechanism was based on mechanical stress induced by the absorption of hydrogen on the Pd film. The reported sensor achieved an average sensitivity of 0.0195 nm/% for a very low concentration of hydrogen i.e., 0.5–1.4%. Since then, several steps have been considered to improve the sensitivity of the sensor by modifying the sensing configurations and alloying the materials. In recent years, some researchers also have developed microstructured fiber-based sensors to overcome the leaching and poor adhesion of Pd, by embedding the Pd nanoparticle along the microstructured hi-birefringence (MHi-Bi) fiber, specifically Panda and Bow-tie as shown in Figure 17 [183]. Standard MHi-Bi fiber can be used for monitoring such distributed stresses in the surrounding [184] as the stress induced in Pd by absorbing hydrogen will change the optical properties of propagated light due to the elasto-optic effect. Exploiting stress variation in hydrogen monitoring has been developed and improved to achieve a good sensing response of the sensors.

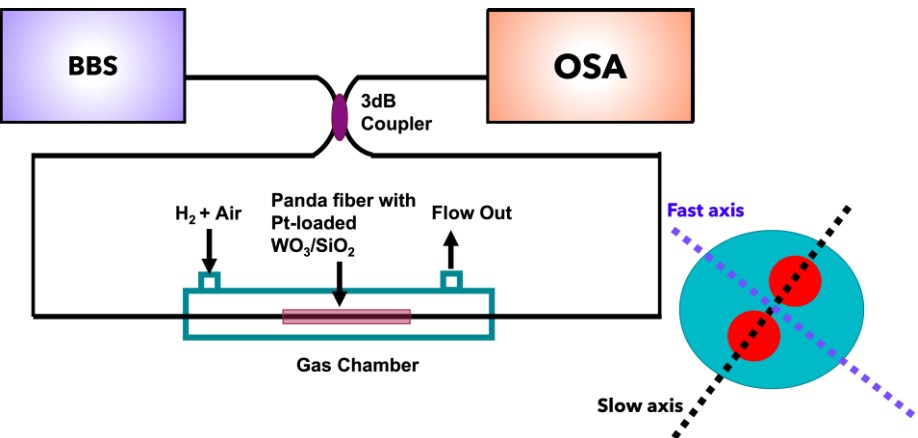

**Figure 17.** Schematic of MHi-Bi fiber with sensing arrangements.

### 4.3. Pd Embedded Hydrogen Sensor

In 2020, Darmadi et al. reported hydrogen sensors fabricated from a thermoplastic nanocomposite material [185]. In the reported work, the author used 3D printing to draw a fiber composed of plasmonically active Pd nanocubes mixed with poly(methyl methacrylate). In the reported work, the sensor responds selectively to hydrogen through a distinct change in its optical properties. Using this technique, the author has achieved a major improvement and, at the same time, the poly (methyl methacrylate) matrix: (i) is a means to stabilize the shape of the embedded nanocrystals; (ii) acts as a molecular sieve that prevents sensor deactivation by other molecular species, such as CO; (iii) prevents their aggregation; and (iv) allow the long-term stability, hence reducing the need of recalibration.

### 5. Future Prospective

Every sensing configuration and approach has its own advantages and limitations. In terms of their sensing performance, these devices can be ranked as SPR > evanescent-field > FBG > MOF, according to their sensing principle and dominated by the configuration's detection accuracy [186]. In terms of the sensing principles, the plasmonic sensors only required a thin metallic film of less than tens of nanometer in order to produce SPR effect, and hence improve the response time from several minute to few seconds. The evanescent field-based hydrogen sensors are identical to SPR, with a thinner sensitive layer and a rapid response. When the light transmits through the etched, tapered, and side polished fiber it generates higher EF at the interface of fiber and sensing layer. The absorption of EF changes corresponding to the variation of the concentration of hydrogen. Therefore, the hydrogen concentration can be determined by measuring the variation in the transmitted light intensity. However, the major limitations of such devices are their fragility with may weaken the sensing head and hence influence its mechanical stability.

Besides, the mechanical strength of the output intensity signal can easily be interrupted by external disturbances, which may lead to poor detection accuracy. However, these challenges were reduced to some extent by using alternative sensing configurations such as grating-based sensors. The grating-based hydrogen sensors are considered as the most promising one with potential applications. On the other hand, fiber grating has an ability of multiplexing using multiple gratings compatible to various wavelengths written on a standard single mode fiber. On the other hand, FBG is an excellent passive optical device for temperature measurement. Hence the effect of ambient temperature can be compensated by adopting reference FBG, which can greatly reduce the deviation caused by temperature. However, the sensitivity of FBG hydrogen sensors is insufficient as well as the LOD, relatively, restricted to principle and demodulation. The sensing performance of these devices can be improved by reducing the diameter of FBG. Reducing the fiber diameter is an effective way to enhance the sensitivity of the hydrogen sensor [187]. Although, the nanofiber is capable of enhancing the sensitivity, however, the excessive reduction of the diameter may lead to decreases in the mechanical stability of such devices [188]. The LPFG based sensor with a higher EF at the cladding may be more promising than FBG based hydrogen sensors. A SAW based hydrogen sensor can be considered as an alternative technology to improve the sensing response without compromising the mechanical stability and response time. The SAW based hydrogen sensor can be coated with Pd alloy to improve the blistering effect, reduce the hysteresis along with rapid response time. Among various alloy formations, PdAu alloy film has been reported to suppress the hysteresis by 17% [51]. Some other works have also shown that the alloying Pd with multiple metals such as (PdAuCu) to overcome several shortcomings simultaneously, such as adhesion, sensitivity, stability, and hysteresis-free response [155]. The Cu was used to protect the sensing material from several toxic gases and improve the stability. Some researcher have also achieved a broad pressure range using Hafnium and Tantalum as hydrogen sensing material [57]. The sensors to be used in marine, aquaculture, agriculture, and similar critical environment have to be robust in nature. The Pd based hydrogen sensor easily gets affected with humidity and other poisonous gasses present in such environment such as CO and NO which can significantly affect the sensing performance of the device. Therefore, an additional protective layer over the sensing material may improve the lifetime of the device. Considering the unstable behavior of Pd in high humidity, Xiangyu et al.[109] utilized two thin 20 μm thick commercial Teflon, to protect the sensing head. The Teflon decreased the effect of $H_2O$ on the Pd coating but allowed the diffusion of hydrogen to the Pd surface. This can be considered as a potential approach to prevent the degradation of sensing material in harsh environment like ocean, agricultural, and aquaculture environment. The optical fiber sensors have the potential to provide early information about the danger of hydrogen leakage. However, the ambient environment of the employed sensor may not be always favorable which includes heat, water, humidity, thunderstorm, etc., which may affect the sensing performance of the device and lead to false or inaccurate data. To protect the sensor and reinforced the performance of the device there are several approaches has been considered by the researchers. As discussed above, researchers utilized polymer (e.g., teflon, polydimethylsiloxane, poly(methyl methacrylate), etc.) as a protective layer and also a sensing membrane to improve hydrogen diffusion and prevent the sensor from humidity. Another author used metal-packaging of fiber sensors which provide a relative strength in strain stability, repeatability, spectra shape, creep, and better temperature response. There are many approach has been utilized previously to reinforce the optical fiber sensor for critical environment and improve the life time and accuracy of the device.

Hydrogen leakage sensors are critical to assure the safe deployment of hydrogen systems; but, because there exists a broad range of sensor options, selecting an appropriate sensor technology can be complicated. Some sensor technologies might not be suitable for a specific application [27]. Nowadays, there are several hydrogen detection sensors commercially available and various new sensing technologies are continued to be developed and can be expected to be commercialized successfully. The most common mature commercial

sensor platforms for hydrogen include electrochemical, evanescent wave, Colorimetric and indicator dyes, and Pd and Pd-alloy films based on various transduction platforms. Depending on the applications, the requirements on sensor performances could vary. As a matter of fact, the U.S. National Renewable Energy Laboratory (NREL) published a workshop report presenting the sensor specifications expected in various domains where hydrogen is used [50,189]. Although discrepancies are observed in the sensor performances specifications according to the targeted application, the Department of Energy (DOE) in the United States provided a short list (shown in Table 6) of target sensor performances to guide sensors developers in order to meet the needs of hydrogen community [190]. The sensor has to satisfy the system safety requirement for it to be commercialized. All the articles we reviewed here fail in some domain to reach the requirement e.g., if the sensor achieves the detection range, it fails to provide accuracy and rapid response time.

**Table 6.** Standard system safety requirements.

| Parameter | System Safety Requirements |
|---|---|
| Detection range | 0.1 to 10 vol% |
| Operating temperature | $-30\,°C$ to $80\,°C$ |
| Gas environment | Ambient air, 10% to 98% relative humidity |
| Accuracy | 5% of full scale |
| Response time | <1 s |
| Life time | 10 years |

## 6. Conclusions

In this article, we aim to provide a comprehensive review of the developments of optical fiber-based hydrogen sensors reported in the past decades. The article covers the latest developments in optical fiber-based hydrogen sensing technology including its sensing configuration and monitoring principles; the sensing response is briefly addressed based on various sensing configurations. Pd and Pd-alloys are shown to play a major role in the reliability of such sensors. Finally, the existing problems and future research directions are also outlined, which were of high relevance for optical fiber hydrogen sensing. Obviously, the experts and early researchers who are interested in this field could not only see the unique feature and flexibilities in the structural design of optical fiber hydrogen sensors, but could also broaden their thoughts and present some new solutions to further exploit more novel optical fiber hydrogen sensors. Additionally, with this review we also aim to give confidence to the general audience for the regular and safe use of hydrogen.

**Author Contributions:** Conceptualization, B.M.A.R.; methodology, B.M.A.R. and A.K.P.; validation, B.M.A.R.; formal analysis, A.K.P., S.V. and N.S.; investigation, A.K.P.; writing—original draft preparation, A.K.P., S.V. and N.S.; writing—review and editing, C.V., R.C. and B.M.A.R.; supervision, C.V., R.C. and B.M.A.R.; funding acquisition, B.M.A.R. All authors have read and agreed to the published version of the manuscript.

**Funding:** This research received no external funding.

**Institutional Review Board Statement:** Not applicable.

**Informed Consent Statement:** Not applicable.

**Data Availability Statement:** Not applicable.

**Acknowledgments:** This work was supported in part by the City, University of London, United Kingdom.

**Conflicts of Interest:** The authors declare no conflict of interest.

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
