# Peer review of "Recent Advances in Optical Hydrogen Sensor including Use of Metal and Metal Alloys: A Review"

_photonics, doi:10.3390/photonics10020122_

Round 1
Reviewer 1 Report
Article is well organized covering detailed study over the sensing materials and various optical systems used for hydrogen sensing. However I find the article can include some more information to reach the standard of Photonics Journal. The comments given below:
1. Author’s are advised to add an graphical abstract in make it more attractive.
2. What are the limitations of WO3? I believe these sensing configuration are also utilized with WO3 and Alloys, why Author cover Pd and Pd alloys only? Why WO3 and Alloy fails to reach the commercialization? Please provide a detailed discussion.
3. The reported article lacks of figures and schematic of configuration making the article less interesting, in my opinion Author should consider taking permissions to reuse figures and add few more figures in each type of optical sensing systems.
4. There are several optical hydrogen systems are reported in recent years however they still fail to reach a commercialize stage. Authors are advised to compare these optical system with commercially available hydrogen sensing systems and why they still lacking behind.
Author Response
#Reviewer-1
Article is well organized covering detailed study over the sensing materials and various optical systems used for hydrogen sensing. However I find the article can include some more information to reach the standard of Photonics Journal. The comments given below:
Ans: All the authors are highly thankful to reviewer’s suggestions. We are truly grateful to your valuable comments and thoughtful suggestions. Based on these comments and suggestions, we have revised this manuscript. Therefore, we have included an itemized list of our responses below. We feel the reviewer made a number of excellent suggestions and we have tried our level best to adopt them and address them. We believe the comments and suggestions have significantly improved the quality of manuscript and made it publishable. For your convenience, please check our point-by-point response letter which we have carefully addressed the points raised by the reviewer. In addition, to facilitate review, we have highlighted significant changes to the text within the revised manuscript using track change.
Q1. Authors are advised to add an graphical abstract in make it more attractive.
Ans: All the authors are thankful to reviewer’s suggestion. We have included a graphical abstract in the revised manuscript as a summary of the review. Graphical abstract included in the revvised manuscript.
Q2. What are the limitations of WO3? I believe these sensing configuration are also utilized with WO3 and Alloys, why Author cover Pd and Pd alloys only? Why WO3 and Alloy fails to reach the commercialization? Please provide a detailed discussion.
Ans: Authors thank the reviewer for his/her suggestion. Considering the suggestion we have added an additional subsection to compare the advantage and limitations of WO3 and Pd along with their possibility of commercialization.
Q3. The reported article lacks of figures and schematic of configuration making the article less interesting, in my opinion Author should consider taking permissions to reuse figures and add few more figures in each type of optical sensing systems.
Ans: All the authors thank again for this suggestion. We have included additional figures in the revised manuscript corresponding to each type of fiber based sensors. The added figures are: Fig. 1, Fig.8, Fig. 11, Fig. 16, and Fig. 17.
Q4. There are several optical hydrogen systems are reported in recent years however they still fail to reach a commercialize stage. Authors are advised to compare these optical system with commercially available hydrogen sensing systems and why they still lacking behind.
Ans. All the authors are thankful to reviewer’s suggestion. We have provided a short paragraph highlighting why these sensors fail to reach commercialization in the revised manuscript. The included part is shown below:
“Hydrogen leakage sensors are critical to assure the safe deployment of hydrogen systems; but, because there exists a broad range of sensor options, selecting an appropriate sensor technology can be complicated. Some sensor technologies might not be suitable for a specific application [27]. Nowadays, there are several hydrogen detection sensors are commercially available and various new sensing technologies are continued to be developed and can be expected to be commercialized successfully. The most common mature commercial sensor platforms for hydrogen includes electrochemical, evanescent wave, Colorimetric and indicator dyes, and Pd and Pd-alloy films based on various transduction platforms. Depending on the applications, the requirements on sensor performances could vary. As a matter of fact, the U.S. National Renewable Energy Laboratory (NREL) published a workshop report presenting the sensor specifications expected in various domains where hydrogen is used [50,189]. Although discrepancies are observed in the sensor performances specifications according to the targeted application, the Department of Energy (DOE) in The United States provided a short list (shown in Table 6) of target sensor performances to guide sensors developers in order to meet the needs of hydrogen community [190]. The sensor has to satisfy the system safety requirement for them to be commercialized. All the article we reviewed here fails in some domain to reach the requirement e.g., if the sensor achieves the detection range it fails to provide accuracy and rapid response time.

Reviewer 2 Report
This review paper is very interesting and I suggest some points to improve.
1- Introduction needs to be broader and add ssome literature about the importance of hydrogen sensing measurement for critical areas like aquaculture, agriculture, marine, etc. Please consider to add: Optics & Laser Technology 140, 107082, 2021; https://pubs.acs.org/doi/10.1021/acssensors.9b01074
2- More details about future perspectives in critical sectors like agriculture, marine and critical platforms.
3 - Reforcing the optical fiber sensors with critical membranes.
Author Response
Reviewer#2
This review paper is very interesting and I suggest some points to improve.
Ans: All the authors are highly thankful to reviewer’s suggestions. We are truly grateful to your critical comments and thoughtful suggestions. Based on these comments and suggestions, we have revised this manuscript. Therefore, we have included an itemized list of our responses below. We feel the reviewer made a number of excellent suggestions and we have tried our level best to adopt them and address them. We believe the comments and suggestions have significantly improved the quality of manuscript and made it publishable. For your convenience, please check our point-by-point response letter which we have carefully addressed the points raised by the reviewer. In addition, to facilitate review, we have highlighted significant changes to the text within the revised manuscript using track change.
Q1.Introduction needs to be broader and add some literature about the importance of hydrogen sensing measurement for critical areas like aquaculture, agriculture, marine, etc. Please consider to add: Optics & Laser Technology 140, 107082, 2021; https://pubs.acs.org/doi/10.1021/acssensors.9b01074
Ans. All the authors are highly thankful to reviewer’s suggestion’s to include the importance of hydrogen in ocean and marine life. We have added a few lines in the introduction and cited the suggested article. The included part is shown below for reviewer’s consideration:
“Hydrogen is also known to be released from the ocean by submarine hydrothermal activity. The dissolved hydrogen has been shown to play an important role in a variety of biological process in the marine environment. For example, hydrogen can be produced and consumed by several aerobic aquatic organisms, and it is an important intermediate in the anaerobic oxidation or organic matter [8–10]. In addition to marine environment the hydrogen is also necessary in aquaculture. When several fish farms do not have access to shore power, then a number of 4 out 10 farms can run on diesel generators which produces CO2 [11]. To fulfill their requirements approximately 10 tons of green hydrogen per plant needs to be produced. The coastal aquaculture industry is considered as the most predominant form of the aquaculture but has caused numerous conflicts among stakeholders because of the limited availability of power in near-coast areas combined with relevant environmental issues [12]. The latter combined with the additional need of power for seafood production has led to a growing trend for the industry to move further away from coast. Sustainable and competitive offshore economic activity though, faces many challenges. A critical one is related to energy requirements and how to provide this vital element uninterruptedly far from coast. The hydrogen energy can play a crucial role for supplying energy offshore. The requirement of hydrogen on large scale required a strategic need for better means of in situ hydrogen monitoring. This drives the need for real time, low-cost, sensitive, and reliable hydrogen leakage detection to give public the confidence in using this technology routinely and more widely.”
Q2. More details about future perspectives in critical sectors like agriculture, marine and critical platforms.
Ans. All the authors are highly thankful to reviewer’s suggestion. We have extended the future prospects in terms of critical environmental conditions. The incorporated part is shown below for reviewer’s consideration:
“The sensors to be used in marine, aquaculture, agriculture and similar critical environment have to be robust in nature. The Pd based hydrogen sensor easily gets affected with humidity and other poisonous gasses present in such environment such as CO and NO which can significantly affect the sensing performance of the device. Therefore an additional protective layer over the sensing material may improve the life time of the device. Considering the unstable behavior of Pd in high humidity Xiangyu et al.[109] utilized two thin 20 μm thick commercial Teflon, to protect the sensing head. The Teflon decreased the effect of H2O on the Pd coating but allows the diffusion of hydrogen to the Pd surface. This can be considered as a potential approach to prevent the degradation of sensing material in harsh environment like ocean, agricultural, and aquaculture environment”
Q3. Reforcing the optical fiber sensors with critical membranes.
Ans. All the authors are highly thankful to reviewer’s suggestion over the reinforcing the fiber sensors. We have included a short paragraph in future prospects section to provide an overview on the reinforcing of optical fiber sensors. The incorporated part is shown below for reviewer’s consideration:
“The optical fiber sensors have the potential to provide early information about danger of hydrogen leakage. However the ambient environment of employed sensor may not be always favourable which includes heat, water, humidity, thunderstorm, etc., which may affect the sensing performance of the device and lead to false or inaccurate data. To protect the sensor and reinforced the performance of the device there are several approaches has been considered by the researchers. As discussed above, researchers utilized polymer (e.g., teflon, polydimethylsiloxane, poly(methyl methacrylate), etc.) as protective layer and also sensing membrane to improve hydrogen diffusion and prevent the sensor from humidity. Some author used metal-packaging of fiber sensors which provide a relative strengths in strain stability, repeatability, spectra shape, creep and better temperature response. There are many approach has been utilized previously to reinforce the optical fiber sensor for critical environment and improve the life time and accuracy of the device.”

Reviewer 3 Report
See attached file

Author Response
Reviewer#3
General Comments
This paper presents a general review of recent advances in optical hydrogen sensing with a particular focus on sensors that employ metals and their alloys. The review is very comprehensive and well structured. Almost all of the relevant papers, that this reviewer is aware of, have been referenced. Examples of a few exceptions will be provided later in this letter, along with a small number of suggested amendments
Ans: All the authors are highly thankful to reviewer’s suggestions. We are truly grateful to your critical comments and thoughtful suggestions. Based on these comments and suggestions, we have revised this manuscript. Therefore, we have included an itemized list of our responses below. We feel the reviewer made a number of excellent suggestions and we have tried our level best to adopt them and address them. We believe the comments and suggestions have significantly improved the quality of manuscript and made it publishable. For your convenience, please check our point-by-point response letter which we have carefully addressed the points raised by the reviewer. In addition, to facilitate review, we have highlighted significant changes to the text within the revised manuscript using track change.
Specfic Comments
First Comment
On line 157 of the manuscript, the authors introduce a relation which can be used to determine the permittivity of Pd which has been exposed to a 4% hydrogen concentration. This equation represents a very early method of modelling PdH permittivity which has been superseded by other methods. The specific limitations of the presented equation is that it fails to account for:
- The wavelength dependent nature of the hydrogenation effect on permittivity.
- The different effects of hydrogenation on the real an imaginary parts of the permittivity.
In view of this, it is recommended that the permittivity model first suggested by Perrotton el al. [1] (last paragraph of section 2) should be introduced here as a superior model. Furthermore, the paper by Downes et al. [2] contains details of a 6th order polynomial fit for the permittivity of Pd exposed to 4% hydrogen which is based on Perrotton’s suggestion (see Equations 10, 11, 12). This reviewer feels the paper by Downes et al. should also be referenced.
Ans: All the authors are highly thankful to reviewer’s suggestion. We have included the limitations of the equation and provide the solution reported by Perrotton et al. and Downes in the revised manuscript. The included part is provided below for reviewer’s consideration:
“The equation 2 can be used to determine the permittivity of Pd exposed to a 4% hydrogen concentration. However, there are some specific limitations of the equation which fails to address (i) the wavelength dependent nature of the hydrogenation effect on permittivity (ii) the different effect of hydrogenation on the real and imaginary parts of the permittivity. These limitations was resolved by Perrotton et al. in 2011 [36] and later modified by Downes et al. in 2017 [37]. Downes et al. determined the dielectric permittivity of PdH by fitting a 6th order polynomial to experimental data generated by Rottkay and Mubin [38]. The polynomial fit covered a broad wavelength range varying from 300 nm to 2000nm. The dielectric permittivity of pure palladium to that of palladium in the presence of 4% hydrogen can be represented by following relation,
(3)
(4)
(5)
where, a and b are the Bessel’s functions.”
Second Comment
The list of referenced papers is very comprehensive, however I feel that the following papers should also be included:
- A paper by Javahiraly [3] which presents a comprehensive review of “fiber optic sensors based on different designs with palladium”.
- A paper by Silva et al. [4] which presents a review of palladium-based fiber-optic sensors.
Ans: All the authors are highly thankful to reviewer’s suggestion. We have included these references in the revised manuscript.
Third Comment
In table 4, the authors reference a paper by Perrotton et al. [1] which details an Ag/SiO2/Pd based sensor. This is a simulation-only paper which doesn’t give details on response time or LOD. However, the same authors subsequently published an experimental study [5] of a sensor based on the same sensing structure within which response time and LOD data are given. It is suggested that this paper also be referenced in table 4.1
It is also suggested that a reference to this “subsequent” paper of Perrotton’s [5] is made on line 515 where the numerical paper by Perrotton [1] is referenced. Some additional explanation should also be included in the text to explain that this additional paper by Perrotton et al. [5] has detailed the response time of the sensor.
Ans: All the authors are highly thankful to reviewer’s suggestions. We have included and cited additional literature reported by Perrotton et al. in the revised manuscript and also included both the articles in Table 5 of the revised manuscript as reference [131] and [36]. The revision is shown below for reviewer’s consideration:
“In 2013, same group validated similar configuration experimentally, but with Au as plasmonic layer [131]. The sensor consists of a multilayer of Au/SiO2/Pd of thickness 35/180/3.75 nm, respectively, coated on a MMF. The sensitivity and selectivity of the device was optimized for 3.75 nm thin Pd layer. The fabricated sensor exhibited good sensitivity to a hydrogen concentration varying from 0.5 to 4% H2 in Ar, along with a good response time of 15 s”
Further to this, another study by Downes et al. [2] presents a simulation-only study of a sensor based on Ag/SiO2/PdY. In this work, palladium has been alloyed with yttrium in order to overcome the de-lamination which Pd is prone to following several hydrogenation/de-hydrogenation cycles. This paper should also be referenced in table 4 and also somewhere in a relevant section of the main text.
Ans: All the authors are highly thankful to reviewer’s suggestion. We have included the article and results reported by Downes et al. in the revised manuscript. The revision is included after Fig. 14 and also cited the same in Table 5 of the revised manuscript as reference [37]. Here we are providing the revision part for reviewer’s consideration:
“In 2016, Downes and Taylor reported a numerical investigation of plasmonic hy-drogen sensor using multilayer capped with an alloy of Pd and Y as the hydrogen sensitive layer [37]. The sensor employs a multilayer stack of Ag/SiO2/PdY of thickness 50.5/72/3 nm, respectively, over the core of a MMF. The PdY alloy was considered to improve the lifetime of the device by reducing the mechanical stress on the sensing layer occurs during sensing operation. In the reported work, the author numerically investigated the performance of the sensor in terms of the sensitivity and detection accuracy for 4% hydrogen concentration. This sensor exhibited an average sensitivity and a detection accuracy of 17.64 nm and 0.014 nm−1, respectively towards the 4% hydrogen concentration. The sensitivity of the device was defined as the shift in resonant wavelength with respect to the overall change in surrounding gaseous hydrogen concentration”
On line 200, this reviewer feels the paper by Downes et al. [2] should be added as an additional “yttrium” reference.
Ans: Authors are thankful to reviewer’s suggestion. We have included the reference in the revised manuscript.
Typographical Issues
- The long paragraph on page 11 could be split into smaller paragraphs to facilitate an easier read.
- Abstract first line change “sensing technology. . . ” to “sensing technologies. . . ”
- Line 58: change “. . . station required. . . ” to “. . . station is required. . . ”
- Line 81: change “approached” to “approaches”
- Line 85: change “Hubert” to “Hübert”
- Line 256: change “. . . configuration, authors. . . ” to “. . . configuration, the authors. . . ”
- Line 360: change “. . . improve. . . ” to “. . . improves. . . ”
Ans: All the authors are highly thankful to reviewer’s suggestion over typographical errors. We have incorporated the suggestion in the revised manuscript and also revised the manuscript thoroughly to make it typo free.
References
- C. Perrotton, N. Javahiraly, M. Slaman, B. Dam, and P. Meyrueis, "Fiber optic surface plasmon resonance sensor based on wavelength modulation for hydrogen sensing," Optics Express 19(S6), A1175–A1183 (2011).
- F. Downes and C. M. Taylor, "Theoretical investigation into the optimisation of an optical fibre surface plasmon resonance hydrogen sensor based on a PdY alloy," Measurement Science and Technology 28(1), (2017).
- N. Javahiraly, "Review on hydrogen leak detection: Comparison between fiber optic sensors based on different designs with palladium," Optical Engineering 54(3), (2015).
- S. F. Silva, L. Coelho, O. Frazao, J. L. Santos, and F. X. Malcata, "A review of palladium-based fiber-optic sensors for molecular hydrogen detection," IEEE Sensors Journal 12(1), 93–102 (2012).
- C. Perrotton, R. Westerwaal, N. Javahiraly, M. Slaman, H. Schreuders, B. Dam, and P. Meyrueis, "A reliable, sensitive and fast optical fiber hydrogen sensor based on surface plasmon resonance," Optics Express 21(1), 382–390 (2013).

Round 2
Reviewer 2 Report
The paper is good for publication